# Calcium sensing by the STIM1 ER-luminal domain

Aparna Gudlur [1], Ana Eliza Zeraik[1,2], Nupura Hirve[1], V. Rajanikanth[1,9], Andrey A. Bobkov[3], Guolin Ma [4], Sisi Zheng [5], Youjun Wang [5], Yubin Zhou [4], Elizabeth A. Komives[6] & Patrick G. Hogan[1,7,8]

Stromal interaction molecule 1 (STIM1) monitors ER-luminal $Ca^{2+}$ levels to maintain cellular $Ca^{2+}$ balance and to support $Ca^{2+}$ signalling. The prevailing view has been that STIM1 senses reduced ER $Ca^{2+}$ through dissociation of bound $Ca^{2+}$ from a single EF-hand site, which triggers a dramatic loss of secondary structure and dimerization of the STIM1 luminal domain. Here we find that the STIM1 luminal domain has 5–6 $Ca^{2+}$-binding sites, that binding at these sites is energetically coupled to binding at the EF-hand site, and that $Ca^{2+}$ dissociation controls a switch to a second structured conformation of the luminal domain rather than protein unfolding. Importantly, the other luminal-domain $Ca^{2+}$-binding sites interact with the EF-hand site to control physiological activation of STIM1 in cells. These findings fundamentally revise our understanding of physiological $Ca^{2+}$ sensing by STIM1, and highlight molecular mechanisms that govern the $Ca^{2+}$ threshold for activation and the steep $Ca^{2+}$ concentration dependence.

[1] Division of Signalling and Gene Expression, La Jolla Institute for Allergy & Immunology, La Jolla, CA 92037, USA. [2] Instituto de Física de São Carlos, Universidade de São Paulo, São Carlos CEP 13563-120 SP, Brazil. [3] Protein Production and Analysis Facility, Sanford Burnham Prebys Medical Discovery Institute, La Jolla, CA 92037, USA. [4] Center for Translational Cancer Research, Institute of Biosciences & Technology, College of Medicine, Texas A&M University, Houston, TX 77030, USA. [5] Beijing Key Laboratory of Gene Resource and Molecular Development, College of Life Sciences, Beijing Normal University, Beijing 100875, China. [6] Department of Chemistry and Biochemistry, University of California–San Diego, La Jolla, CA 92037, USA. [7] Program in Immunology, University of California–San Diego, La Jolla, CA 92037, USA. [8] Moores Cancer Center, University of California–San Diego, La Jolla, CA 92093, USA. [9] Present address: H Lee Moffitt Cancer Center & Research Institute, Tampa, FL 33612, USA. Correspondence and requests for materials should be addressed to P.G.H. (email: phogan@lji.org)

Stromal interaction molecule 1 (STIM1), an ER-membrane protein, is a pivotal regulator of cellular Ca$^{2+}$ balance and Ca$^{2+}$ signalling. Each STIM1 monomer consists of an ER-luminal domain specialized for Ca$^{2+}$ sensing in the range ~100–400 μM, a single transmembrane helix, and a cytoplasmic domain capable of regulated interaction with plasma membrane ORAI Ca$^{2+}$ channels at ER-plasma membrane contacts (Fig. 1a)[1–3].

Early mechanistic investigations into how STIM1 controls Ca$^{2+}$ signalling focused on the single predicted EF-hand in the STIM1 luminal domain. Mutations that compromised Ca$^{2+}$ binding to the EF-hand led to constitutive STIM1 activation, evidenced by the relocalization of STIM1 to ER-plasma membrane junctions in resting cells and by constitutive Ca$^{2+}$ influx[4–7]. An NMR structure of the luminal region STIM1(58-201) with Ca$^{2+}$ bound revealed a compact domain consisting of paired EF-hands—one canonical, the second noncanonical and unable to bind Ca$^{2+}$— closely interacting with a sterile α motif (SAM) domain[8]. STIM1(58-201) is termed the 'EFSAM' domain on this basis. Two other elements with clearly defined functions, both located in the STIM1 cytoplasmic region, are the STIM1 ORAI activating region (SOAR; residues 344-442) or CRAC activation domain (CAD; residues 342-448) that interacts with and gates the ORAI channel[9–12]; and the C-terminal polybasic tail (residues 671-685) that targets active STIM1 to ER-plasma membrane junctions[9,11,13,14].

The STIM1 cytoplasmic domain undergoes an inactive > active transition in which availability of the SOAR/CAD domain and the polybasic tail is regulated by the level of free Ca$^{2+}$ in ER stores. Inactive STIM1 in cells is a dimer held together by SOAR/CAD–SOAR/CAD interactions[15–19]. The SOAR/CAD domain is folded back against CC1 in a way that precludes its effective interaction with plasma membrane ORAI channels[19,20]. STIM1 activation is triggered when Ca$^{2+}$ dissociates from STIM1 ER-luminal domains, the EFSAM domains rearrange or dimerize, and the STIM1 cytoplasmic domains undergo a conformational change that releases the SOAR/CAD domains and the polybasic tails[8,19,21–27]. Biochemical and biophysical studies have established that STIM communicates the store-depletion signal to the cytoplasm via STIM–STIM association in the transmembrane and predicted coiled coil 1 (CC1) regions[19,26–28]. These intradimer interactions stabilize a coiled coil and bury the CC1 surface that would otherwise retain SOAR/CAD near the ER in resting cells. Active STIM1 relocalizes to ER-plasma membrane junctions, and its extended cytoplasmic domain bridges the ~ 15 nm ER-plasma membrane distance, and recruits and gates ORAI channels[3].

The accepted description of how STIM1 senses reduced ER Ca$^{2+}$ has come from studies of the isolated recombinant STIM1 luminal domain. The triggering event has been thought to be Ca$^{2+}$ dissociation from canonical EF-hand sites, leading in turn to a major loss of EFSAM secondary structure and dimerization of EFSAM domains[8,21,22], and apposition of TM-CC1 regions[19,26–28]. Here we find that Ca$^{2+}$ sensing entails loss of Ca$^{2+}$ from both the EF-hand and other sites in EFSAM; that the additional binding sites interact with the EF-hand site to set the Ca$^{2+}$ sensitivity of STIM1; and that luminal domain unfolding is not a prerequisite for the EFSAM-EFSAM dimerization that drives STIM1 activation.

## Results

### A soluble STIM1 construct for study of Ca$^{2+}$ binding.

The EFSAM domain of STIM1 has been studied previously as an isolated domain to parse the STIM1 Ca$^{2+}$ response. In cells, the EFSAM domain is linked via the STIM1 transmembrane helix and CC1 region to dimeric SOAR/CAD, in both the STIM1 inactive and active conformations (Fig. 1a). We reasoned that the

EFSAM conformational transitions that take place upon Ca$^{2+}$ binding and dissociation might be replicated more faithfully in experiments where EFSAM was similarly linked to a dimeric protein than in experiments with isolated EFSAM. Therefore, we undertook to design a fully soluble protein, omitting the transmembrane segment, and incorporating a stably dimerizing C-terminal domain. This more natural linkage might incidentally reduce the nonphysiological aggregate formation characteristic of the isolated EFSAM domain at low concentrations of Ca$^{2+}$. After trials with candidate dimers of known and stable structure, we settled on an EFSAM protein coupled through a short linker to *Thermus thermophilus* GrpE (Fig. 1b, c). EFSAM-GrpE was soluble when expressed in bacteria, unlike the isolated EFSAM domain, which needed to be purified under denaturing conditions and refolded[21,24]. EFSAM-GrpE showed no change in migration on size-exclusion chromatography in the presence or absence of Ca$^{2+}$ (Fig. 1d), and, importantly for the intended use, the purified protein did not aggregate in the absence of Ca$^{2+}$.

### Ca$^{2+}$-dependent conformational change in EFSAM-GrpE.

A characteristic early indicator of Ca$^{2+}$ dissociation from the STIM1 luminal domain in cells is STIM–STIM FRET between N-terminal fluorescent protein labels. We designed a FRET experiment to test for similar sensing of Ca$^{2+}$ by EFSAM-GrpE in vitro. EFSAM-GrpE dimers were randomly labelled with fluorescein and Alexa Fluor 594 at an engineered N-terminal cysteine in the EFSAM domain (Fig. 1e, Supplementary Fig. 1a). As with CFP/YFP labels in cells, an appreciable fraction of EFSAM-GrpE dimers will contain donor–donor or acceptor–acceptor pairs, and with chemical labelling some sites will remain unlabelled, so in the best case only half of the dimers can exhibit intradimer FRET. Samples rigorously depleted of Ca$^{2+}$ by passage over Chelex resin exhibited FRET (Fig. 1f), indicating close apposition of the labels in the two EFSAM domains. The observed FRET was between labels in the same EFSAM-GrpE dimer, since a mixture of comparable amounts of singly donor-labelled and acceptor-labelled proteins exhibited no FRET (Supplementary Fig. 1b). Further, FRET was reduced as increasing concentrations of Ca$^{2+}$ were added (Fig. 1g), showing that in vitro, as in cells[9,16,29], Ca$^{2+}$ causes a relative movement of the EFSAM domains.

To verify that the change in FRET upon Ca$^{2+}$ addition was not an isolated finding that reflected the particular EFSAM fusion construct used, we replicated the experiment with EFSAM-SAH-GrpE, a construct in which EFSAM was connected to GrpE by a monomeric single α-helix linker the length of CC1 (Supplementary Fig. 1c–h). The results were similar, with this longer construct also showing substantial FRET in the absence of Ca$^{2+}$, and reduced FRET in the presence of Ca$^{2+}$. Thus EFSAM-GrpE replicates a defining aspect of the Ca$^{2+}$-dependent STIM1 conformational change.

Notably, in both cases, the midpoint of the transition to lower FRET falls at ~1–10 μM Ca$^{2+}$, suggesting that at least one Ca$^{2+}$ is bound to EFSAM with $K_d$ below ~ 10 μM. This value differs from the $K_d$ of a STIM1 EF-hand grafted into CD2 domain 1 (~500 μM; ref. [30]). The most plausible explanation is that Ca$^{2+}$ binding to the EF-hand in the full EFSAM domain is stabilized allosterically by the interaction of the EF-hand portion with the SAM domain, much as Ca$^{2+}$ binding to calmodulin can be stabilized by interaction with target proteins or peptides[31–33]. The value measured in the EFSAM-GrpE FRET assay also differs from the measured IC$_{50}$ for inhibition by Ca$^{2+}$ of STIM1 activation in cells ( ~ 200 μM; refs. [34,35]) and from that determined in our Ca$^{2+}$ titration of STIM1(A230C) crosslinking[28]. The higher IC$_{50}$ characteristic of full-length STIM1 is most likely to reflect an energetic cost exacted by the coupling between Ca$^{2+}$ binding to

the STIM1 EFSAM domain and the concerted conformational change of full-length STIM1 from the active to the inactive form.

The key experiments defining STIM1 function in cells, including the observation of STIM–STIM FRET upon $Ca^{2+}$ store depletion[9] and measurement of the $Ca^{2+}$ concentration dependence of STIM activation[34,35], were conducted at room temperature. We have carried out this initial FRET experiment and other experiments in this paper under comparable conditions. In a series of parallel experiments, we have seen no qualitative difference in STIM1 relocalization in cells responding to store depletion at room temperature and at 37 C.

**$Ca^{2+}$-dependent conformational change in cells**. It has been unclear how much of the increase in STIM1-STIM1 FRET during store depletion is due to EFSAM–EFSAM rearrangement and how much may arise from SOAR/CAD oligomerization (see discussion in ref. [3]). To examine the contribution of EFSAM rearrangements to STIM–STIM FRET in cells, we engineered EFSAM-TM-CC1-GrpE constructs extending to residue 343 of CC1, fused to GrpE in place of SOAR/CAD, and labelled at the N terminus with either CFP or YFP (Fig. 2a–d). HeLa cells coexpressing the CFP- and YFP-labelled proteins exhibited a FRET increase upon store depletion with thapsigargin (TG) that was comparable in magnitude to that of cells expressing CFP- and YFP-labelled full-length STIM1 (Fig. 2a, b, d). A tandem CFP-YFP fusion protein expressed as a control exhibited higher FRET, presumably reflecting the higher fraction of donor–acceptor dimers, but, as expected, showed no response to store depletion (Fig. 2c, d). As there is no evidence from our in vitro gel filtration experiments for higher-order oligomerization of EFSAM-GrpE (Fig. 1d), the results suggest that intradimer rearrangement of EFSAM domains accounts for an appreciable part of the STIM1–STIM1 FRET change in cells.

**Isothermal titration calorimetry detects multiple $Ca^{2+}$ sites**. Isothermal titration calorimetry (ITC) measures the heat released or absorbed in a chemical reaction following sequential injections of one of the reactants, and thus permits quantitative studies of ligand binding. To assess $Ca^{2+}$ binding to EFSAM-GrpE over the range of physiological ER-luminal $Ca^{2+}$ concentrations, we titrated $Ca^{2+}$ into an initially $Ca^{2+}$-free EFSAM-GrpE sample, with a corresponding titration into initially $Ca^{2+}$-free GrpE serving as control (Fig. 3a, b). The plot of heat released in the EFSAM-GrpE titration leads to three immediate conclusions. First, from the range of molar ratios of $Ca^{2+}$:protein in which heat release is observed, there are ~5 $Ca^{2+}$-binding sites per EFSAM monomer (see discussion in Methods). Second, the sites are $Ca^{2+}$-specific, since similar $Ca^{2+}$ binding is observed in the presence of 2 mM $Mg^{2+}$ (Supplementary Fig. 2). Third, considering the amounts of $Ca^{2+}$ added in the titration, the sites in wild-type EFSAM are occupied at concentrations of free $Ca^{2+}$ below 1 mM (Fig. 3a, legend). It is also evident that the titration is biphasic. In light of the FRET data reported above (Fig. 1g), it is likely that net heat release early in the $Ca^{2+}$ titration is the sum of an exothermic $Ca^{2+}$ binding component and an endothermic EFSAM conformational change component. The latter interpretation is backed by ITC measurements, below, on the EFSAM-2NQ mutant, which does not undergo a conformational change and which exhibits a monophasic exothermic binding reaction. These qualitative conclusions are independent of any specific model that might be used to derive binding parameters from the calorimetry data (see discussion in Methods).

An EF-hand mutation might have been expected to result in the loss of one $Ca^{2+}$-binding site per monomer, but in fact EFSAM(D76A)-GrpE exhibited no $Ca^{2+}$ binding detectable by

ITC (Fig. 3c). This result indicates that $Ca^{2+}$ binding at the EF-hand is a prerequisite for $Ca^{2+}$ binding at the other sites detected in wild-type EFSAM. A corollary is that previous experiments in which the D76A replacement and other replacements in the EF-hand led to loss of the STIM1 $Ca^{2+}$ response[4–7] do not exclude physiologically relevant $Ca^{2+}$ binding at additional sites.

In simpler cases, ITC data can be analyzed to parse ligand binding to individual sites. Here, given the interdependence of binding at the EF-hand and the other $Ca^{2+}$-binding sites, the coupling of $Ca^{2+}$ binding to a change in intradimer EFSAM–EFSAM interactions, and the large number of parameters that would be required for a realistic description of binding, this more detailed analysis is not feasible. However, we have been successful in determining the concentration range in which $Ca^{2+}$ binds to EFSAM with two alternative approaches described below, a competitive binding assay presented in the next section and a fluorescence enhancement assay using the environment-sensitive dye 8-anilino-1-naphthalenesulfonic acid (ANS).

**A fluorescence assay confirms multiple $Ca^{2+}$ sites in EFSAM**. We sought to confirm the existence of multiple $Ca^{2+}$-binding sites in the STIM1 luminal domain by an independent technique (Fig. 3d–g). We chose a competitive binding experiment because it would also illuminate a blind spot of ITC—the ITC titration is designed to use concentrations of protein in excess of $K_d$ and does not directly report free $Ca^{2+}$ at any point in the titration. The rationale of the competitive binding experiment (Fig. 3d) is that the fluorescent $Ca^{2+}$ sensor D4 at trace levels can report free $Ca^{2+}$ but, due to its low concentration in the incubation (150 nM here), will not appreciably alter free $Ca^{2+}$. A $Ca^{2+}$-binding competitor protein, in this case EFSAM, that is present at a fixed concentration will bind a fraction of added $Ca^{2+}$ at each step in the titration and shift the titration curve. D4 surveys $Ca^{2+}$ concentrations in the range from tens to hundreds of μM (ref. [36]), and was chosen as the fluorescent $Ca^{2+}$ sensor based on the ITC measurements of Fig. 3a.

In a control titration including 35 μM GrpE, binding of $Ca^{2+}$ to the D4 sensor was fitted with $K_d$ 200 μM (Fig. 3e), in line with the published value for D4 alone[36]. Inclusion of EFSAM(D76A)-GrpE at 35 μM caused no shift in the titration curve, indicating no detectable binding of $Ca^{2+}$ to the D76A protein (Fig. 3f). Inclusion of wild-type EFSAM-GrpE at 35 μM resulted in clear competition, and the competition could not be accounted for by a single $Ca^{2+}$-binding site (Fig. 3g, Supplementary Fig. 3). We estimated free $Ca^{2+}$ at each step in the titration from the D4 signal, and $Ca^{2+}$ bound to EFSAM as the difference between total $Ca^{2+}$ added and free $Ca^{2+}$ (Fig. 3h). The resulting binding relation for EFSAM is plotted in Fig. 3i. The main conclusion is that ~6 sites per monomer are occupied in the range of $Ca^{2+}$ studied. Occupancy of most of these sites (5–6 sites) takes place in the range from 50–400 μM $Ca^{2+}$, suggesting that the sites are physiologically relevant for ER-luminal $Ca^{2+}$ binding.

**A surface implicated in $Ca^{2+}$ binding and STIM relocalization**. There are 21 aspartate or glutamate residues in the NMR structure of STIM1 EFSAM domain that are neither EF-hand $Ca^{2+}$ ligands nor in close contact with other parts of the protein. We replaced aspartate with asparagine (D > N) and glutamate with glutamine (E > Q)— conservative and sterically similar replacements that neutralize the negative charge— in three spatially separate groupings of these EFSAM acidic residues (Fig. 4a), on the premise that function would be affected if the substituted region contributes to $Ca^{2+}$ binding and stabilizes inactive STIM1.

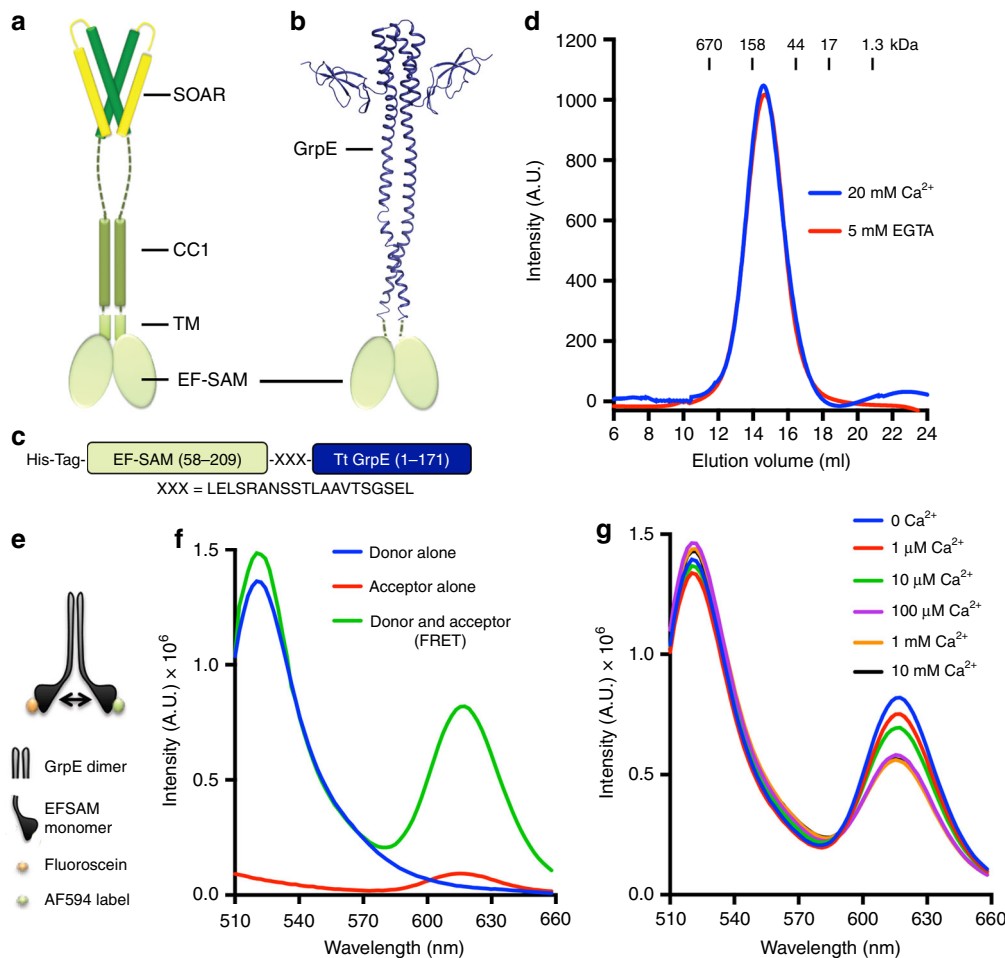

**Fig. 1** EFSAM-GrpE design and Ca$^{2+}$ responsiveness. **a** Cartoon of activated STIM1 (58–473) as inferred from the literature[45]. Domain organization is marked. Residues 24–57 and 473–685 are not depicted. **b** Cartoon of the expected EFSAM-GrpE structure used in the current study, showing structural similarity to the extended activated STIM1. Green ovals represent EFSAM (58-209) and blue cartoon denotes *Thermus thermophilus* GrpE. GrpE is not structurally related to STIM1 except for the presence of extended α-helices that form a coiled coil. The coiled coil is constitutive in GrpE, unlike in STIM1. **c** Schematic of the EFSAM-GrpE construct design. **d** Size exclusion chromatography of the Ca$^{2+}$-bound (20 mM Ca$^{2+}$; blue line) and Ca$^{2+}$-free (5 mM EGTA; red line) forms of EFSAM-GrpE. **e** Schematic of chemical labelling of EFSAM-GrpE, depicting the case where individual monomers are labelled with fluorescein and AF594. Other possible combinations in the random labelling approach used here are not illustrated. **f** Fluorescence emission spectra ($\lambda_{ex} = 420$ nm) of single-cysteine EFSAM-GrpE labelled with fluorescein (Donor alone), AF594 (Acceptor alone) or with both dyes (Donor and Acceptor). All measurements were made in Ca$^{2+}$-free buffer. **g** Fluorescence spectral scans ($\lambda_{ex} = 420$ nm) of double-labelled EFSAM-GrpE with varying amounts of Ca$^{2+}$

The three altered STIM1 proteins were designated STIM1-1NQ, STIM1-2NQ, and STIM1-3NQ.

STIM1-1NQ and STIM1-3NQ displayed an ER-like pattern in unstimulated cells and relocalized to ER-plasma membrane junctions after store depletion (Fig. 4b). These results do not support a major role for region 1 or region 3 acidic residues in stabilizing the inactive conformation of STIM1.

STIM1-2NQ, unexpectedly, showed little or no relocalization after store depletion (Fig. 4b, Supplementary Note 1). This effect is opposite to the expected effect of a simple loss of stabilizing Ca$^{2+}$-binding sites, and Supplementary Figs. 4, 5, 6, and 7, and Supplementary Note 2 provide compelling evidence that reducing Ca$^{2+}$ failed to trigger an activating conformational change in the STIM1-2NQ cytoplasmic domain. The D > N and E > Q substitutions in EFSAM-2NQ did, in fact, alter Ca$^{2+}$ binding markedly, since ITC showed that only a single Ca$^{2+}$-binding site was occupied in the concentration range monitored (Fig. 4c). We return later to the question whether a less extensive set of replacements in region 2 can separate the effect on Ca$^{2+}$ binding from the negative effect on STIM1 activation.

**Loss of Ca$^{2+}$ has little impact on EFSAM secondary structure.** It seemed likely that the 2NQ mutations interfered either with the EFSAM conformational change or with the EFSAM–EFSAM dimer interaction underlying activation. We focused on the conformational change by examining the 2NQ and wild-type EFSAM-GrpE fusion proteins using far-UV circular dichroism (CD) spectroscopy, which reports on secondary structure and, in particular, on α-helix content. The α-helix content of both EFSAM-2NQ-GrpE and wild-type EFSAM-GrpE in the presence of Ca$^{2+}$ was in line with the NMR structure of the STIM1 EFSAM domain (Fig. 5a, b)[8], and in fact the secondary structure of EFSAM-2NQ was indifferent to the presence or absence of Ca$^{2+}$ (Fig. 5b), in line with our hypothesis. However, these experiments produced the completely unexpected finding that wild-type EFSAM-GrpE exhibited only a minor loss of secondary structure in the absence of Ca$^{2+}$ (Fig. 5a), not the massive unfolding that has been observed with the isolated EFSAM domain.

To verify that we could observe a loss of EFSAM secondary structure by CD if it occurred, we followed the thermal denaturation of EFSAM-GrpE. EFSAM-GrpE showed a clear

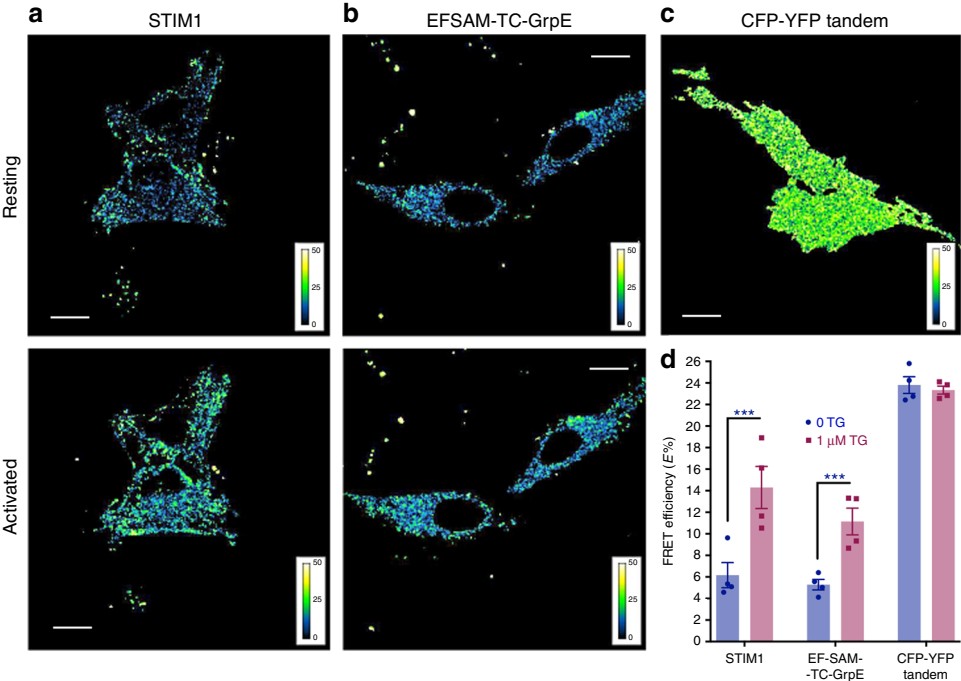

**Fig. 2** $Ca^{2+}$-dependent conformational change in cells. **a–b** Confocal micrographs showing FRET before (resting; upper panels) and after 1 μM TG stimulation (activated; lower panels) in representative HeLa cells co-expressing CFP- and YFP-tagged full-length STIM1 (**a**; $n = 7$) or EFSAM-TC-GrpE (**b**; TC = STIM1 transmembrane and CC1 regions; $n = 7$). Scale bars, 5 μm. False-color scale representing normalized FRET efficiency, expNFRET, for each image is included as inset. **c** Confocal micrograph showing FRET in HeLa cells expressing a CFP-YFP tandem construct as a control, without TG treatment ($n = 5$). Scale bar, 5 μM. False-color scale representing normalized FRET efficiency, expNFRET, is included as inset. **d** Quantification of FRET changes in **a–c**. Error bars report SEM. CFP-YFP FRET change in HeLa cells expressing either STIM1 or EFSAM-TC-GrpE was significant after treatment with 1 μM TG (marked with ***, $p < 0.05$ by an unpaired, two-tailed Student's $t$-test)

decrease in α-helical secondary structure between 25 °C and 65 °C (Supplementary Fig. 8). In contrast, GrpE showed only a minimal decline in α-helical structure in this temperature range, in keeping with published evidence that the main unfolding transition of GrpE occurs above 90 C (ref [37]).

The EF-hand mutation D76A compromises STIM1 $Ca^{2+}$ binding and thus provides an independent way to examine the effect of $Ca^{2+}$ dissociation on STIM1 protein structure. The far-UV CD spectrum of EFSAM(D76A)-GrpE was comparable to the spectrum of the wild-type protein in the absence of $Ca^{2+}$, reflecting a largely α-helical structure, and was unchanged by addition of $Ca^{2+}$ (Fig. 5c). Thus any loss of helical structure in the $Ca^{2+}$-free EFSAM domain is minor, and $Ca^{2+}$ sensing reflects a $Ca^{2+}$-dependent conformational rearrangement between two folded protein structures.

**H-D exchange shows that $Ca^{2+}$-free EFSAM is structured.** Hydrogen-deuterium exchange–mass spectrometry (HDX–MS) is a sensitive way to probe local structure and dynamics of the protein backbone. A standard protocol is to expose the protein briefly to $D_2O$, then minimize further H-D exchange in a low-pH buffer, digest with pepsin, and measure incorporation of deuterium into identified peptic peptides by LC-MS. Deuterium is rapidly incorporated at backbone –NH groups in unstructured regions of the protein. On the other hand, backbone sites that are hydrogen bonded or inaccessible to solvent are protected from rapid H-D exchange. Protection is not absolute, but depends on the time scale probed, and hence the time course of exchange yields additional information on local backbone dynamics.

We subjected wild-type EFSAM-GrpE to deuterium exchange for periods ranging from 0.5–5 min, in the presence of 30 μM $Ca^{2+}$ and in the absence of $Ca^{2+}$, and assessed deuterium

incorporation (Supplementary Fig. 9a). $Ca^{2+}$-bound EFSAM exhibited substantial protection from exchange, consistent with its known folded structure (Fig. 5d–h). $Ca^{2+}$-free EFSAM showed altered deuterium uptake throughout the EF-hand and SAM domains, but uptake was altered in an unexpected direction: the backbone –NH groups were detectably more protected from exchange in the absence of $Ca^{2+}$ across nearly the entire EFSAM sequence (Fig. 5h, Supplementary Fig. 9b). Only a single short section, the peptide [136]EVIQWLIT[143], stood out for its contrasting pronounced protection in 30 μM $Ca^{2+}$ (Fig. 5f, h). In counterpoint, peptides flanking this section— [121]WKAWKSSE-VYMWTVDE[136] and, at 5 min, [144]YVELPQYEET[153]— exhibited a greater increase in H-D exchange in the presence of $Ca^{2+}$ than the remainder of the protein (Fig. 5e, h, Supplementary Fig. 9b).

HDX–MS data for EFSAM(D76A) and EFSAM-2NQ buttressed the evidence that there are two structured states of EFSAM. H-D exchange in these proteins was almost $Ca^{2+}$-insensitive, and, broadly speaking, each mutant protein mirrored one state of wild-type EFSAM: Exchange in EFSAM(D76A) was similar to exchange in wild-type EFSAM in the absence of $Ca^{2+}$, and, conversely, exchange in EFSAM-2NQ largely matched exchange of wild-type EFSAM in the presence of 30 μM $Ca^{2+}$ (Fig. 5h, Supplementary Fig. 9b–d, Supplementary Note 3).

Measurements with intrinsic and extrinsic fluorescent probes (Supplementary Fig. 10, Supplementary Note 4) further support the conclusion from the CD and HDX–MS experiments that there are structured conformations of EFSAM at both endpoints of the high $Ca^{2+}$–low $Ca^{2+}$ conformational change.

**EFSAM does not unfold in full-length STIM1.** To determine whether EFSAM in full-length STIM undergoes obligatory unfolding when it transitions to the active dimeric form, we

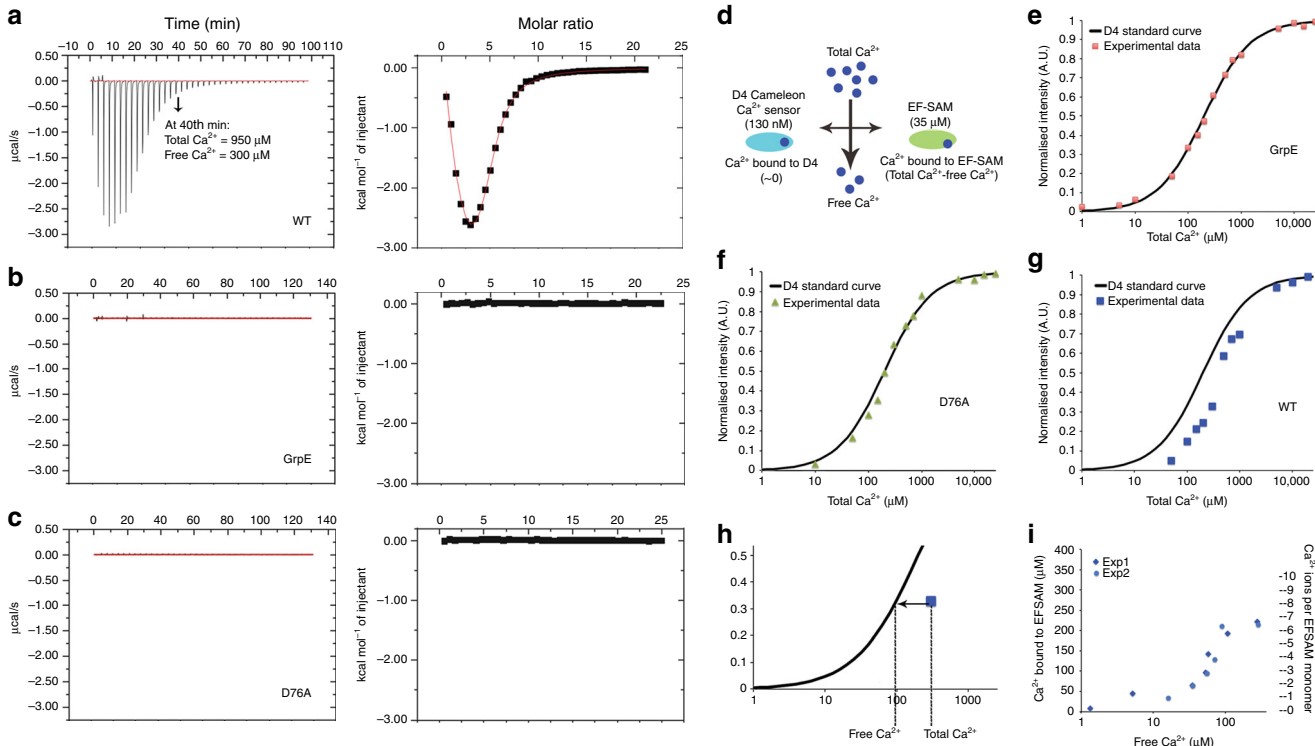

**Fig. 3** Multiple Ca$^{2+}$-binding sites in EFSAM. **a–c** Isothermal titration calorimetric (ITC) analyses of EFSAM-GrpE WT (**a**), GrpE (**b**), and EFSAM-GrpE D76A (**c**). Left panels, Heat changes measured by injecting 1 μl aliquots of 13 mM Ca$^{2+}$ into a sample cell containing initially 130 μM protein, stated as the monomer concentration. Right panels, Integrated binding isotherm as a function of molar ratio (Ca$^{2+}$: protein) after subtracting heats of dilution. In **a**, the indicated total Ca$^{2+}$ at 40 min is the amount added to the ITC cell up to that point, corrected for the increase in volume, and free Ca$^{2+}$ is calculated as total Ca$^{2+}$ minus bound Ca$^{2+}$ assuming five sites are occupied in EFSAM. **d** D4 sensor Ca$^{2+}$ competition assay design. **e–g** D4 sensor normalized fluorescence signals in the presence of GrpE (35 μM) (**e**), EFSAM(D76A)-GrpE (35 μM) (**f**), or WT EFSAM-GrpE (35 μM) (**g**) are compared to the Ca$^{2+}$-binding curve of D4 alone (black solid trace in each panel). **h** Method used for calculating free Ca$^{2+}$ and Ca$^{2+}$ bound to EFSAM (as total Ca$^{2+}$ minus free Ca$^{2+}$) from the competition data in **g**. **i** Plot of Ca$^{2+}$ bound to WT EFSAM as a function of free Ca$^{2+}$ concentration, calculated as in **h**. Data from two experiments plotted individually

probed whether cysteine residues engineered at sites that are buried or partially buried in Ca$^{2+}$-bound EFSAM-GrpE become exposed upon dissociation of Ca$^{2+}$ from full-length STIM1 in ER membranes. We engineered a panel of constructs appropriate for the experiment, and carried out the experiment in parallel with full-length STIM1 in ER membranes and with EFSAM-GrpE. Except for one M > C replacement, the engineered proteins had relatively conservative V > C, T > C, and S > C replacements at sites that are buried or partially buried in the NMR structure. All the mutant STIM proteins (expressed as GFP-STIM1) responded to depletion of ER Ca$^{2+}$ stores by forming STIM1 puncta (Supplementary Fig. 11). The S126C, V129C, and V198C mutants—and to a lesser extent the A168C mutant— had noticeable STIM1 fluorescence remaining in the deeper ER after store depletion, suggesting that these four mutants might be less strongly activated by store depletion.

We prepared HeLa cell membranes in the same way as for STIM1-STIM1 crosslinking experiments, where we have shown that STIM1 undergoes a conformational change in response to Ca$^{2+}$ (ref. [28] and this manuscript). We incubated the membranes in buffer with 0.5 mM EGTA or 2 mM Ca$^{2+}$, probed exposure of the individual engineered cysteine residues by reaction with biotin-maleimide, then captured biotinylated proteins on streptavidin resin, followed by SDS-polyacrylamide gel electrophoresis and western blotting for STIM1. The residues tested showed similar low accessibility in the presence of Ca$^{2+}$ for full-length STIM1 and for EFSAM-GrpE (Fig. 6, Supplementary Figs. 11 and 12a). The exposed S58C control was well labelled in both proteins, and the

buried cysteine residues in EFSAM-GrpE became exposed and could be labelled upon denaturation with 6 M urea. However, only minor changes in accessibility of the individual buried residues were seen in the two proteins when comparing the Ca$^{2+}$ and EGTA conditions. Thus, the EFSAM domain of full-length STIM1 did not unfold upon dissociation of Ca$^{2+}$.

Exposure of residues S126C, V129C, and V137C appeared somewhat higher in 2 mM Ca$^{2+}$ than in EGTA, reminiscent of the greater deuterium exchange into the peptide [121]WKAWKS-SEVYMWTVDE[136] in the presence of Ca$^{2+}$, but further detailed experiments will be needed to determine whether this indicates local unfolding.

**EFSAM surface residues tune STIM1 Ca$^{2+}$ sensitivity.** We returned to the question whether EFSAM Ca$^{2+}$ binding can be perturbed without preventing STIM1 activation. Examining a series of STIM1 proteins mutated at a subset of the 2NQ sites for altered resting localization of GFP-STIM1 led to the discovery of STIM1-2NQ(6–11), a protein with N/Q substitutions only at E94, D100, E111, D112, E118, and D119. STIM1-2NQ(6–11) was partially constitutively activated, as evidenced by the presence of some puncta already in resting cells, and by the prominent appearance of puncta upon brief exposure to Ca$^{2+}$-free solution (Fig. 7a). The latter 'hair-trigger' response was not characteristic of cells expressing wild-type STIM1 (compare Fig. 7a, b, middle panels), and presumably arose from a heightened sensitivity to the limited store depletion that occurred during a brief period in

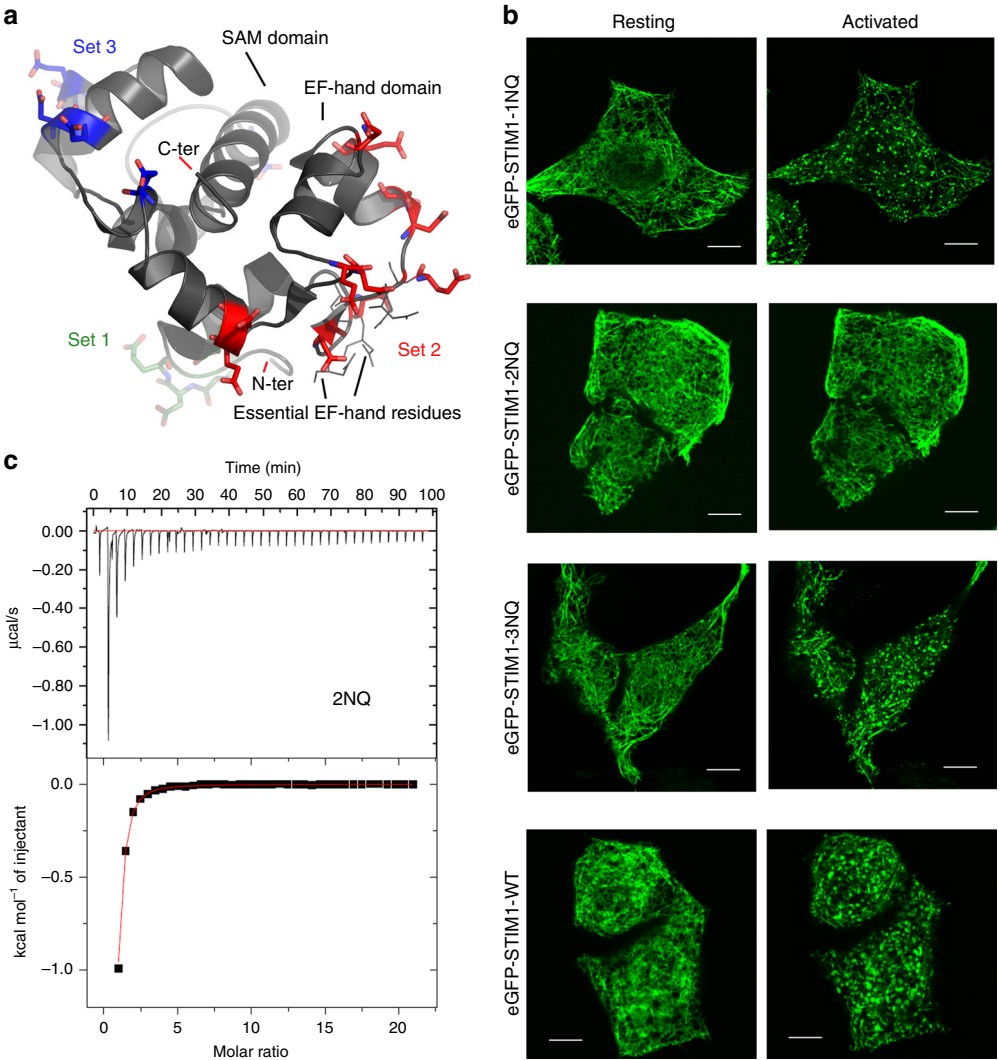

**Fig. 4** Identification of an EFSAM surface that regulates STIM1 $Ca^{2+}$ binding and puncta formation. **a** Three sets of surface residues (stick representation) studied in this work, marked on the cartoon diagram of EFSAM structure (PDB ID: 2K60). Region 1 comprises four acidic residues near the EFSAM N terminus, region 2 comprises eleven acidic residues spanning the exposed surface of the two EF-hands, and region 3 comprises six residues on a surface of the SAM domain. **b** Confocal micrographs of HeLa cells expressing eGFP-STIM1-1NQ, eGFP-STIM1-2NQ, eGFP-STIM1-3NQ, or wild-type eGFP-STIM1, at rest (left panels) and after store depletion with 1 μM thapsigargin (TG) (right panels). Scale bars, 5 μm. **c** ITC analysis of EFSAM-2NQ. Top panel, heat changes measured by injecting 1 μl aliquots of 25 mM $Ca^{2+}$ into a sample cell containing initially 252 μM protein. Bottom panel, integrated binding isotherm as a function of molar ratio ($Ca^{2+}$: protein) after subtracting heats of dilution

$Ca^{2+}$-free solution, due to $Ca^{2+}$ extrusion unbalanced by resting $Ca^{2+}$ influx. Reflecting the partial constitutive activation, cells expressing STIM1-2NQ(6–11) showed constitutive $Ca^{2+}$ entry through ORAI channels (Fig. 7c).

In vitro experiments established the basis of the partial constitutive activation. ITC still reported ~5–6 $Ca^{2+}$-binding sites in EFSAM-2NQ(6–11), but the heat released upon $Ca^{2+}$ binding to EFSAM-2NQ(6–11) was reduced compared to that with wild-type EFSAM (Fig. 7d; compare Fig. 3a), indicating that $Ca^{2+}$ binding was perturbed. Heat release is not in itself a measure of the free energy change ΔG or of binding affinity, but a larger total amount of $Ca^{2+}$ was injected to reach the endpoint of the EFSAM-2NQ(6-11) titration curve in the ITC experiments even though protein concentration in Fig. 7d was comparable to that of wild-type protein in Fig. 3a, suggesting that the binding was of lower affinity. To test this hypothesis, we employed a direct assay of STIM1 conformational change. Oxidative crosslinking at residue A230C in a cysteineless STIM1 background measures apposition of the STIM1 transmembrane helices[28]. In ER

membranes with 'wildtype' STIM1(A230C), the $Ca^{2+}$ concentration dependence of crosslinking[28] mirrors the concentration dependence of STIM1 activation in cells[34,35], whereas the permanently active STIM1(D76A) protein crosslinks regardless of ambient $Ca^{2+}$ concentration[28]. Tellingly, in this assay, the $Ca^{2+}$ dependence of the conformational change of STIM1(A230C)-2NQ(6–11) in isolated ER membranes was shifted rightward into or beyond the range of resting ER $Ca^{2+}$ concentrations (Fig. 7e, f and Supplementary Fig. 12b). As in the case of STIM1-2NQ, we can draw no certain conclusion whether the replaced residues are themselves $Ca^{2+}$ ligands or whether they affect $Ca^{2+}$ binding at separate EFSAM surface sites allosterically. The central point is that the 2NQ(6–11) replacements both alter STIM1 $Ca^{2+}$ binding at EFSAM surface sites and shift the $Ca^{2+}$ concentration dependence of STIM1 activation in vitro and in cells.

## Discussion

We designed the dimeric EFSAM-GrpE protein to incorporate some of the geometric constraints imposed on the STIM1 luminal

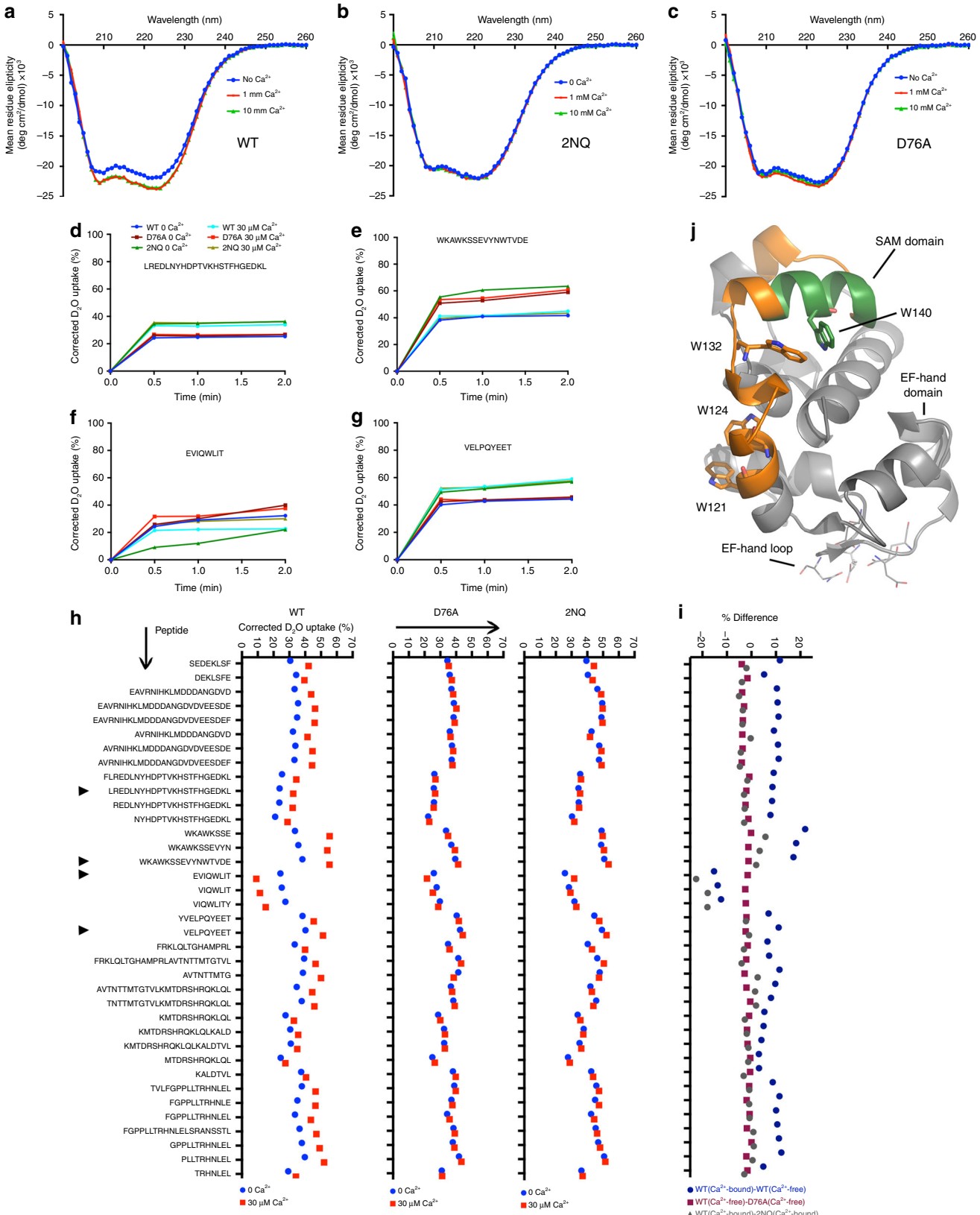

domains in cells by their linkage to the STIM1 transmembrane helices and the dimeric STIM1 cytoplasmic domain. We show here that $Ca^{2+}$ dissociation from EFSAM-GrpE or from full-length STIM1 controls a switch between two structured conformations of the EFSAM domain; that $Ca^{2+}$ at physiological concentrations occupies multiple sites in the EFSAM domain, not solely the EF-hand site; and that the additional EFSAM $Ca^{2+}$-binding sites are required together with the EF-hand site to control physiological activation of STIM1 in cells. We show further that $Ca^{2+}$ binding at the EF-hand and $Ca^{2+}$ binding at

**Fig. 5** Limited change in EFSAM secondary structure after Ca$^{2+}$ depletion. **a–c** Far-UV CD spectra of WT EFSAM-GrpE, EFSAM-2NQ-GrpE, and EFSAM (D76A)-GrpE, each protein 5 µM, in the absence or presence of Ca$^{2+}$. **d–g** Examples of deuterium uptake into four peptic peptides of EFSAM WT, D76A and 2NQ in Ca$^{2+}$-free and Ca$^{2+}$-bound forms, at 0.5, 1, 2 and 5 min time points, determined by HDX–MS. **h** Deuterium exchange into EFSAM peptides of wild-type, D76A, and 2NQ proteins, in 0 Ca$^{2+}$ or 30 µM Ca$^{2+}$, at the 0.5 min time point. The peptides from **d–g** are marked with arrowheads. Supplementary Fig. 9b presents the data for all time points. **i** Differences in deuterium exchange between the inactive and active forms of wild-type EFSAM (WT(Ca$^{2+}$-bound)–WT(Ca$^{2+}$-free)), between the active form of wild-type EFSAM and the D76A protein (WT(Ca$^{2+}$-free)–D76A(Ca$^{2+}$-free)), and between the inactive form of wild-type EFSAM and the 2NQ protein (WT(Ca$^{2+}$-bound)–2NQ(Ca$^{2+}$-bound)), at the 0.5 min time point. Data are the averages of triplicate measurements. **j** Spatial positions of the low-exchanging peptide (EVIQWLIT; green) and two flanking higher-exchanging peptides (WKAWKSSEVYNWTVDE and YVELPQYEET; orange) highlighted on the Ca$^{2+}$-bound WT EFSAM structure (PDB ID: 2K60)

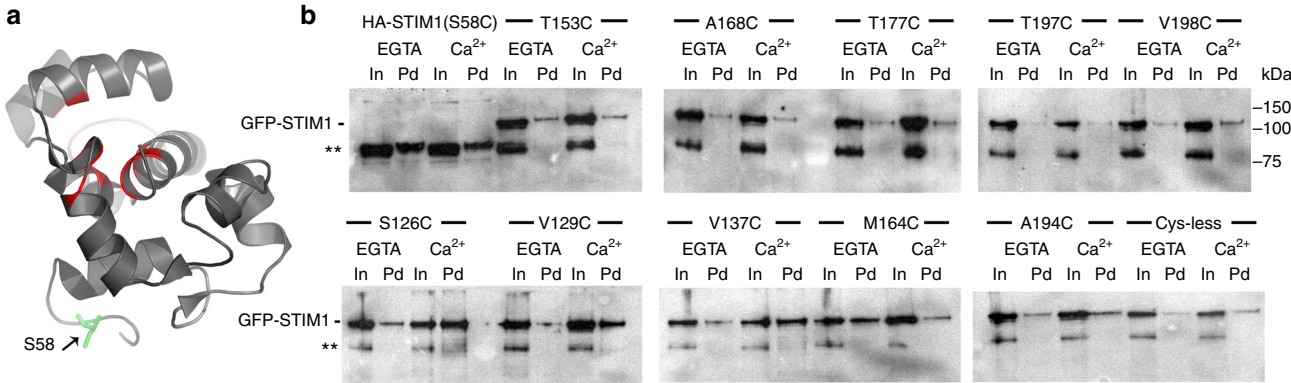

**Fig. 6** Exposure of buried EFSAM cysteine residues in STIM1 is insensitive to Ca$^{2+}$ depletion. **a** Ribbon structure of Ca$^{2+}$-bound EFSAM (PDB ID: 2K60). Red highlighting indicates the placement of engineered cysteine residues to probe possible EFSAM unfolding. Cysteine replacement of the fully exposed residue S58 was a positive control. **b** Anti-STIM1 western blots of cysteine-substituted HA-tagged STIM1 or GFP-STIM1 proteins. Proteins were labelled in HeLa cell membranes by incubation with biotin-maleimide, in the presence of EGTA or Ca$^{2+}$ as indicated, then solubilized, bound to streptavidin, and eluted in SDS sample buffer. Each construct and condition is represented by a pair of lanes: input (In), 2.5% of the material applied to the streptavidin beads; and pulled down (Pd), all material recovered from the beads. The exposed-residue control, HA-tagged STIM1(S58C) in the first four lanes of the upper left gel, was amply biotinylated (Pd lanes). Endogenous STIM1 migrates with HA-STIM1(S58C) and contributes to anti-STIM1 staining of the HA-STIM1(S58C) input lanes. However, it does not contribute to the signal from biotinylated protein (Pd lanes), since endogenous STIM1 detected as the lower band (\*\*) in all other samples (In lanes) was not biotinylated (Pd lanes) in those samples. For the cysteine-substituted GFP-STIM1 samples, detection of biotinylated protein (Pd lanes) in the upper band is the parameter of interest. The uncropped blots are shown in Supplementary Fig. 12

the EFSAM surface sites are energetically coupled, as evidenced by the fact that binding to the EF-hand enables or stabilizes binding to the other sites (Fig. 3a–c). The coupling in the reverse direction has not been examined experimentally, but it is required by thermodynamic considerations (Fig. 8). Our findings compel us to revisit the mechanism at the heart of STIM1 Ca$^{2+}$ sensing.

A starting point for this analysis is the Ca$^{2+}$-dependent conformational change observed with both EFSAM-GrpE and full-length STIM1. The local rearrangement of EFSAM upon Ca$^{2+}$ binding is likely to be similar in EFSAM-GrpE and full-length STIM1, but full-length STIM1 couples the local rearrangement to a concerted conformational change of the entire protein[19,26–28]. We have proposed above that the energetic cost associated with the concerted conformational change explains why the titration curve for the FRET signal of EFSAM-GrpE or EFSAM-SAH-GrpE (Fig. 1g, Supplementary Fig. 1g) is centred at 1–10 µM Ca$^{2+}$, whereas the titrations for STIM1 translocation to ER-plasma membrane junctions[34,35] and for disulfide crosslinking of STIM1 transmembrane segments in ER membranes[28] are centred at ~200 µM Ca$^{2+}$. The challenges are to relate the conformational change to Ca$^{2+}$ binding given that there are now two classes of Ca$^{2+}$-binding sites to consider, and to explain why altering Ca$^{2+}$ binding at surface sites affects the concentration dependence of the conformational change.

A straightforward interpretation retains the assumption that STIM1 relocalization in cells and STIM1–STIM1 crosslinking in

membranes are elicited by dissociation of Ca$^{2+}$ from the EF-hand site, and hence that half-maximal dissociation of Ca$^{2+}$ from the EF-hand site occurs at ~200 µM Ca$^{2+}$. The additional elements that must be incorporated in light of experiments reported here are that the EFSAM surface sites bind Ca$^{2+}$ in that same concentration range (Fig. 3a, i, Supplementary Fig. 10d), that Ca$^{2+}$ binding at the EFSAM surface sites stabilizes binding at the EF-hand site, and that the EFSAM surface sites are coupled to the EF-hand site in a way that requires Ca$^{2+}$ dissociation from the EFSAM surface sites prior to STIM1 activation (Fig. 9). With this interpretation, the activation curve of the STIM1-2NQ(6-11) mutant is shifted to the right because higher Ca$^{2+}$ concentrations are needed for occupancy of the EFSAM surface sites, and therefore, due to energetic coupling, for binding to the EF-hand site.

A second more radical interpretation posits that Ca$^{2+}$ in fact binds to the EF-hand site in full-length STIM1 at low micromolar concentrations, just as it does in EFSAM-GrpE or EFSAM-SAH-GrpE (Fig. 1g, Supplementary Fig. 1g), but that EF-hand site occupancy alone is not sufficient to stabilize full-length STIM1 in its inactive conformation. In this case, STIM1 relocalization in cells and STIM1–STIM1 crosslinking in membranes would be controlled mainly by Ca$^{2+}$ binding to and dissociation from EFSAM surface sites, and the shifted Ca$^{2+}$ sensitivity of STIM1-2NQ(6-11) would be accounted for by the altered binding at those sites without invoking the coupling to

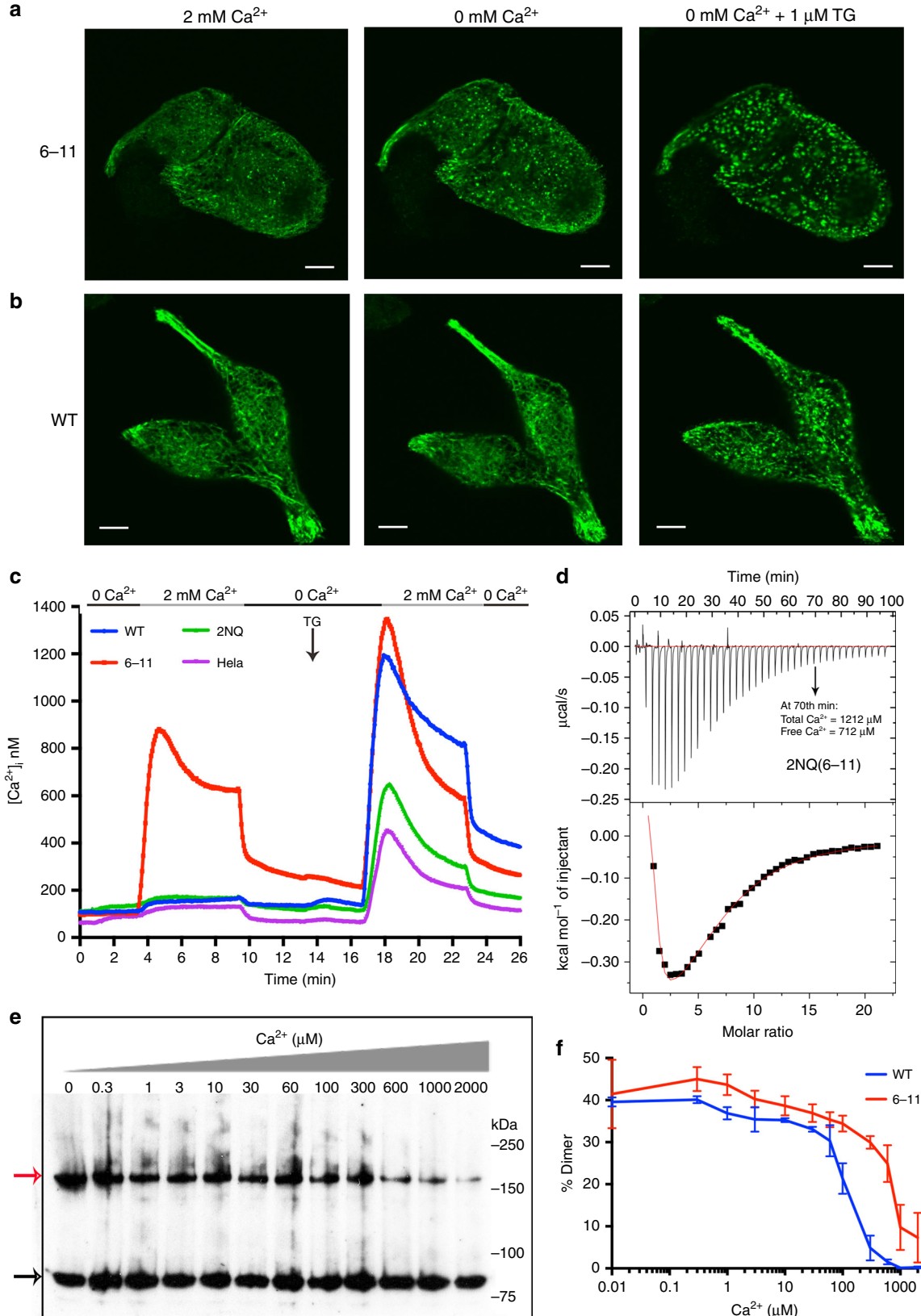

the EF-hand site. Measurements of $Ca^{2+}$ binding to full-length STIM1 in cells or in ER membranes will be necessary to distinguish between these scenarios. Such measurements could also address the possibility, admissible under either scenario, that $Ca^{2+}$ binding at some of the EFSAM surface sites is further

stabilized as part of an EFSAM-protein or EFSAM-lipid interface.

The prevailing model for STIM1 $Ca^{2+}$ sensing has been built on studies of the isolated recombinant EFSAM domain[8,21–24]. Isolated EFSAM differs from EFSAM-GrpE in two key respects: It

**Fig. 7** Effect of 2NQ(6-11) mutations on EFSAM Ca$^{2+}$ sensing. **a–b** Confocal micrographs of GFP-STIM1-2NQ(6-11) (6-11; **a**) and GFP-STIM1 (WT; **b**) at rest (2 mM Ca$^{2+}$; left panels), during a 10-min exposure to Ca$^{2+}$-free solution (0 mM Ca$^{2+}$; middle panels), and following store depletion with 1 μM TG (right panels). Scale bars, 5 μm. **c** Single-cell [Ca$^{2+}$]$_i$ measurements in HeLa cells expressing mCherry-ORAI1 and wild-type GFP-STIM1 (WT; $n = 220$), GFP-STIM1-2NQ(6-11) (6-11; $n = 258$), or GFP-STIM1-2NQ (2NQ; $n = 193$), or in non-transfected HeLa cells (HeLa; $n = 220$). Cells were exposed to solutions containing varied concentrations of CaCl$_2$ or 1 μM thapsigargin (TG) as indicated. **d** ITC analysis of EFSAM-2NQ(6-11). Top panel, heat changes measured by injecting 1 μl aliquots of 10 mM Ca$^{2+}$ into a sample cell containing initially 100 μM protein. Bottom panel, integrated binding isotherm as a function of molar ratio (Ca$^{2+}$: protein) after subtracting heats of dilution. Total Ca$^{2+}$ at 70 min is the amount added to the ITC cell up to that point, corrected for the increase in volume, and free Ca$^{2+}$ is calculated as total Ca$^{2+}$ minus bound Ca$^{2+}$ assuming five sites are occupied in EFSAM. **e** Western blot showing crosslinking of STIM1(A230C)-2NQ(6-11) in cellular membranes incubated at the specified Ca$^{2+}$ concentrations. The black arrow marks the STIM1 monomer band, and the red arrow marks the dimer band. The uncropped blot is shown in Supplementary Fig. 12. **f** STIM-STIM dimer formation for STIM1 (A230C)-2NQ(6-11) at each Ca$^{2+}$ concentration is compared to the corresponding dimer formation for STIM1(A230C). Data from three independent experiments are plotted as mean ± SEM

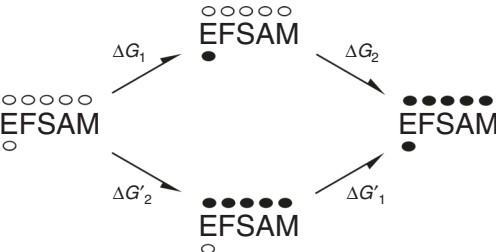

**Fig. 8** Energetic coupling of the EF-hand site and the EFSAM surface sites. Equilibrium diagram for four selected states of an idealized EFSAM protein at a fixed Ca$^{2+}$ concentration. (States with intermediate levels of Ca$^{2+}$ binding that would be present in equilibrium with those represented are considered below.) For each state represented, five surface Ca$^{2+}$-binding sites are indicated above EFSAM and the EF-hand site is indicated below, with bound Ca$^{2+}$ depicted by the filled ellipses. Note that 'EFSAM' denotes the protein, not one particular protein conformation. For example, Ca$^{2+}$-free EFSAM (at left) and EFSAM with Ca$^{2+}$ bound to the EF-hand (at top) would differ in conformation. The $\Delta G$ parameters—free energy differences between pairs of states— are functions of Ca$^{2+}$ concentration. At a given Ca$^{2+}$ concentration, though, each $\Delta G$ value is uniquely defined, and $\Delta G_1 + \Delta G_2 = \Delta G_1' + \Delta G_2'$ (6). The individual $\Delta G$ values and their Ca$^{2+}$ dependence need not be determined in order to understand the energetic coupling. We make only the assumption that the poor binding of Ca$^{2+}$ at physiological concentrations to the surface sites of EFSAM(D76A) reflects comparable poor binding to wild-type EFSAM when Ca$^{2+}$ is not present in the EF-hand site. Then binding at the EF-hand favours binding at the other sites collectively ($\Delta G_2 < \Delta G_2'$), and equation (6) requires that collective binding at the EFSAM surface sites favours binding at the EF-hand ($\Delta G_1' < \Delta G_1$). Notice that an analogous argument can be made for any state having one, two, three, or four surface Ca$^{2+}$ sites occupied, as long as that state is present at an appreciable concentration in wild-type EFSAM at some Ca$^{2+}$ concentration in the physiological range. This logic supports the broad conclusion that binding at the EFSAM surface sites favours binding at the EF-hand site

has a propensity to unfold at low Ca$^{2+}$ concentrations, and it exhibits negligible Ca$^{2+}$ binding at 10 μM Ca$^{2+}$, whether binding is measured directly as $^{45}$Ca$^{2+}$ binding or indirectly as conformational stabilization. The unfolding simply reflects poor structural stability of the isolated EFSAM domain in the absence of Ca$^{2+}$. The inability of Ca$^{2+}$ to bind and stabilize isolated EFSAM at concentrations that elicit a conformational change of EFSAM-GrpE could have same root cause—unfolding of isolated recombinant EFSAM is energetically favored at low Ca$^{2+}$ concentrations, and thus EF-hand Ca$^{2+}$ binding is opposed by the unfavorable free energy change of refolding the protein.

Early quantitative studies of Ca$^{2+}$ binding to isolated EFSAM found a single $^{45}$Ca$^{2+}$-binding site per monomer[8,21]. Furukawa et al. subsequently reported that an isolated EFSAM domain,

purified and refolded under different conditions, exhibited a secondary-structure change that was cooperative with respect to Ca$^{2+}$ concentration, with Hill coefficient 4.7 (ref. [24]). This finding might have led to a re-evaluation of the number of Ca$^{2+}$-binding sites in EFSAM, but Furukawa et al. instead attempted to explain the cooperativity on the basis of a single Ca$^{2+}$-binding site per monomer, and did not determine the number of binding sites experimentally. It seems more likely in light of our current data that the Furukawa EFSAM preparation had multiple Ca$^{2+}$-binding sites, and that the structural change reflected cooperative binding at the EF-hand and surface Ca$^{2+}$-binding sites. The stoichiometry of physiological Ca$^{2+}$ binding is central to understanding STIM1 Ca$^{2+}$ sensing, because the overall process of Ca$^{2+}$ dissociation, STIM1 oligomerization, and STIM1 redistribution in cells is a steep function of ER Ca$^{2+}$ concentration, with Hill coefficient in the range 4–8 (refs. [34,35]). Based on the premise that there is only a single Ca$^{2+}$-binding site per monomer, STIM1 oligomerization has been thought to play a dominant role in this cooperativity. Our results indicate that it is time to reconsider this interpretation.

Our analysis of Ca$^{2+}$ binding to EFSAM-GrpE and the associated protein conformational changes offers a fresh perspective on physiological Ca$^{2+}$ sensing by STIM1. The next obvious steps are to define the additional Ca$^{2+}$-binding sites structurally, and to investigate the extent to which Ca$^{2+}$ dissociation from the multiple sites contributes to the cooperativity of STIM1 activation. Perhaps most intriguingly, the finding that EFSAM does not unfold upon dissociation of Ca$^{2+}$ will spur renewed efforts to define the structure of the EFSAM–EFSAM dimer that triggers STIM1 activation.

## Methods

**Plasmids for bacterial and mammalian expression.** *Thermus thermophilus* HB8 GrpE genetic sequence was synthesized commercially (Genewiz), based on sequence information from PDB (ID: 3A6M; GenBank nucleotide sequence: AB012390.1 (2222–2755 bp)). The synthesized GrpE, nucleotides 2222-2734, was PCR amplified and cloned between SacI and NotI sites of pET28a vector (Qiagen) to generate the His$_6$-tagged GrpE expression construct. The cDNA encoding mouse EFSAM sequence (STIM1 residues 58-209) was PCR amplified and cloned at the NdeI and SacI sites of pET28a vector at the 5′ end of the GrpE sequence for the His$_6$-tagged EFSAM-GrpE expression clone, resulting in a construct encoding EFSAM(58-209)–LELSRANSSTLAAVTSGSEL–GrpE(1-171). The D76A mutation was introduced into the pET28-EFSAM-GrpE plasmid using the QuikChange site-directed mutagenesis kit (Agilent). The single-cysteine EFSAM constructs used for energy transfer experiments were generated by PCR amplification of EFSAM-GrpE, introducing nine nucleotides encoding the protein sequence –CGG– immediately N-terminal to EFSAM residue 58, and cloned into pET28a vector. The 100-residue Single Alpha Helix (SAH) construct (pBIEX1-SAH) from *Sus scrofa* myosin VI is a kind gift from Dr. Sivaraj Sivaramakrishnan, University of Minnesota. The SAH domain was introduced between EFSAM and GrpE sequences using BstBI and AvrII to generate pET28-EFSAM-SAH-GrpE. Expression constructs encoding EFSAM mutated in the surface sites of regions 1, 2, and 3 were generated by mutagenesis of pET28-EFSAM-GrpE plasmid using a commercial service (Genewiz). The 1NQ variant of EFSAM-GrpE contains the mutations E59Q, D60N, E61Q, and E66Q; the 11 mutations of the 2NQ variant are D77N, D82N, E86Q, D89N, E90Q, E94Q, D100N, E111Q, D112N, E118Q, and D119N;

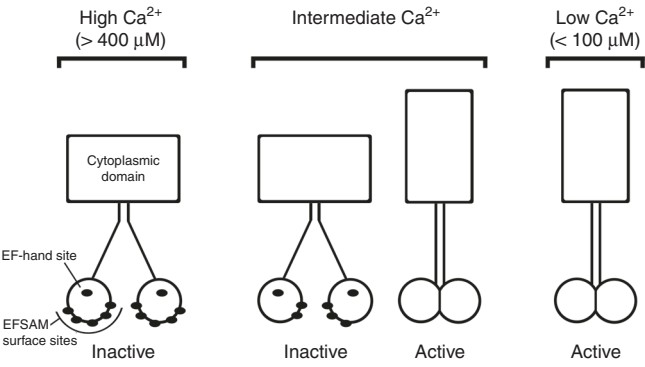

**Fig. 9** Relation between $Ca^{2+}$ binding and STIM1 conformational change in cells. Schematic depiction of $Ca^{2+}$ (black ellipses) bound to EFSAM, in the context of STIM1 dimers, to illustrate the progressive changes in STIM1 that accompany depletion of ER $Ca^{2+}$ stores. The model retains the assumption that $Ca^{2+}$ binding at the EF-hand controls the conformational change. At high levels of ER $Ca^{2+}$, all EFSAM $Ca^{2+}$-binding sites are occupied, and EFSAM is inactive. At low levels of $Ca^{2+}$, no sites are occupied, and EFSAM is active. (The relevant values of ER free $Ca^{2+}$ are taken from the literature[34, 35].) At intermediate $Ca^{2+}$, there is a mixture of STIM1 dimers, with some resembling the high-$Ca^{2+}$ case but having fewer $Ca^{2+}$ bound to EFSAM surface sites, and others having no $Ca^{2+}$ bound. Only a representative 'inactive' dimer is illustrated here. The fact that there are dimers with other levels of surface-site occupancy, and possibly mixed STIM1 dimers with the EF-hand site occupied in only one monomer, would not alter the argument materially. The salient point—taught by EFSAM (D76A) in Fig. 3c, f— is that there will be no EFSAM domain with the EF-hand site empty and surface sites occupied. Stated simply, $Ca^{2+}$ must dissociate both from the EFSAM surface sites and from the EF-hand during activation

and the six mutations of the 3NQ variant are E128Q, D135N, E136Q, E151Q, E152Q, and D183N. The single-cysteine EFSAM-GrpE mutants used for biotinylation experiments were generated by mutagenesis of the pET28-EFSAM-GrpE plasmid using the QuikChange site-directed mutagenesis kit (Agilent).

The HA-tagged STIM1 construct and eGFP-STIM1 are described in detail in ref [28]. Two restriction sites, BsiWI at the 5' end and MluI at the 3' end of the mutated regions, were introduced into the pET28-EFSAM-GrpE to facilitate subcloning of the surface variants. Segments of DNA encoding the surface mutations were spliced into the mammalian vectors using PCR amplification and standard subcloning steps. The single-cysteine eGFP-STIM1 mutants used for biotinylation experiments in cellular membranes were generated by mutagenesis of the eGFP-STIM1 plasmid using the QuikChange site-directed mutagenesis kit (Agilent).

The cDNA encoding STIM2(1-434) was PCR amplified and cloned into the pEGFP-N1 vector using XhoI and BamHI sites to generate STIM2(1-434)-WT-GFP. Subsequently, the D167A mutation was introduced into the STIM2(1-434)-WT-GFP plasmid using the QuikChange site-directed mutagenesis kit (Agilent). The STIM2(1-434)-2NQ-GFP insert was synthesized using commercial services (Genewiz) and encodes the following ten mutations: D168N, E177Q, D180N, E181Q, E185Q, D191N, E202Q, D203N, E209Q and D210N. pET28-STIM2-EFSAM-GrpE was generated by cloning the cDNA corresponding to STIM2(149-299) between NdeI and SacI sites of the pET28-EFSAM-GrpE plasmid.

The pEX-SP-CFP/YFP-STIM1(23-685) constructs used for live-cell FRET measurements were a kind gift from Dr. Tobias Meyer, Stanford University. pEX-SP-CFP/YFP-EFSAM-TC-GrpE plasmids were made by subcloning PCR-amplified fragments of EFSAM-TM-CC1 (encoding human STIM1 residues 58-343) and GrpE into KpnI-AgeI and AgeI-NotI sites, respectively, at the 5' ends of the CFP/YFP coding sequences in place of the full-length STIM1 inserts.

The oligonucleotide primers utilized for PCR amplifications and mutagenesis are listed in Supplementary Table 1.

**Protein purification**. Plasmids encoding EFSAM-GrpE variants were transformed into *E. coli* strain BL21 (NEB) for efficient protein expression. The transformed cells were grown at 37 °C. Protein expression was induced by addition of 1 mM isopropyl-β-D-thiogalactopyranoside (IPTG) when $OD_{600}$ of the culture reached 0.6, followed by incubation for 12 h at 16 °C. Harvested cells were resuspended in lysis buffer containing 50 mM Tris pH 7.5, 150 mM NaCl, 20 mM $CaCl_2$, 5% (vol/

vol) glycerol, 1 mM β-mecaptoethanol, and Roche protease inhibitors (1 tablet per 50 ml buffer), and sonicated. Cell debris was removed by centrifugation, and the lysate was applied to $Ni^{2+}$-nitrilotriacetic acid (Ni-NTA)-agarose resin (Qiagen). Bound recombinant proteins were eluted in lysis buffer containing 250 mM imidazole. Proteins were further purified by gel filtration on a Superdex 200 column (GE Healthcare) in the presence of 20 mM $CaCl_2$, 5 mM EGTA, or Chelex-treated $Ca^{2+}$-free buffer (Chelex-100 resin, Bio-Rad), with 2 mM TCEP (tris(2-carbox-yethyl)phosphine). Eluted proteins were concentrated using Amicon Ultra 30 kDa concentration devices (EMD Millipore).

**Protein conjugation with fluorescent dyes**. Single-cysteine-containing EFSAM-GrpE variants were labelled either singly with fluorescein-5-maleimide (Thermo) or Alexa Fluor 594 $C_5$ maleimide (Thermo), or with both fluorophores, for in vitro energy transfer measurements. Each protein sample, ~100 μM in 50 mM Tris pH 7.5, 150 mM NaCl, was first treated with 10 mM TCEP for at least 1 h to reduce any disulfide bonds. Excess TCEP was removed by passing the fully reduced protein sample through an Amicon 30 kDa device with a quick buffer exchange using 50 mM Tris pH 7.5, 150 mM NaCl buffer. Dye working stocks (10–20 mM) were prepared in anhydrous DMF immediately prior to use. A 10-fold molar excess of the fluorescent dyes was added slowly to the protein and incubated at 4 °C for 1 h. The labelled protein was passed through a PD-10 desalting column (GE Health-care) and further concentrated. Labelled proteins yielded a ~ 0.8-0.9:1 molar ratio of fluorophore to protein, based on measuring the absorbance at 495 nm (fluor-escein-5-maleimide, $\varepsilon_{495} = 68,000$ $M^{-1}$ $cm^{-1}$), 588 nm (Alexa Fluor 594 $C_5$ mal-eimide, $\varepsilon_{588} = 96,000$ $M^{-1}$ $cm^{-1}$), and 280 nm (protein, $\varepsilon_{280} = 34,950$ $M^{-1}$ $cm^{-1}$). Dye conjugation to each protein was further confirmed by resolving the labelled proteins on a 4–12% NuPAGE gel (Life Technologies) and imaging under a broad-range UV light using Gel Doc EZ (Bio-Rad).

**Fluorescence spectrometry-based assays**. For in vitro FRET measurements, fluorescence spectra were acquired at 22 °C using a QuantaMaster 40 spectro-fluorometer (PTI). Protein concentration was 250 nM. The spectra were collected from 450 to 650 nm with the excitation set at fluorescein dye excitation wavelength (420 nm) and the slit widths set at 12 nm.

In the ANS dye-binding assay, 1.4 μM of the specified EFSAM-GrpE variant was mixed with 40 μM ANS (8-anilino-1-naphthalenesulfonic acid) dye (Thermo) in a Chelex-treated buffer containing 50 mM Tris-Cl pH 7.5, 150 mM NaCl, and 5% glycerol. The dye-mixed protein was incubated for 5 min before each spectral scan ($\lambda_{ex} = 370$ nm) with the specified amounts of $CaCl_2$ at 22 °C in dark. The slit width was set at 8 nm and the fluorescence emission spectra were acquired from 440 to 600 nm.

Tryptophan emission spectra ($\lambda_{ex} = 295$ nm) were recorded from 310 to 420 nm for 1 μM wild-type EFSAM-GrpE and its variants, at 4 °C. The slit widths were set at 2 nm. Proteins incubated with 6 M GdCl were used to collect the denatured protein emission spectra.

D4 cameleon sensor expression vector, pBAD-D4, was a gift from Amy Palmer and Roger Tsien (Addgene plasmid #37473; ref. [36]). 150 nM D4 sensor, resuspended in Chelex-treated buffer containing 50 mM Tris-Cl pH 7.5, 150 mM NaCl and 5% glycerol, was titrated with specified amounts of $Ca^{2+}$, in the presence or absence of 35 μM EFSAM-GrpE variants. Emission spectra were recorded from 450 to 600 nm at $\lambda_{ex} = 410$ nm with 2 nm slit width, at 22 °C.

In the D4 fluorescence competition assay, $Ca^{2+}$ binding to D4 is proportional to the normalized intensity change $y$. Data points plotted in Fig. 3e–g were normalized as

$$y = (y_{OBS} - y_{MIN})/(y_{MAX} - y_{MIN}), \qquad (1)$$

where $y_{MIN}$ is the intensity measured in $Ca^{2+}$-free solution, and $y_{MAX}$ the intensity in 25 mM $Ca^{2+}$. The D4 standard binding curve in Fig. 3e–g is for binding to a single site with $K_d$ 200 μM,

$$y = [Ca^{2+}]/(K_d + [Ca^{2+}]). \qquad (2)$$

In Fig. 3i, data points from two experiments are plotted individually. Data in the individual experiments were normalized as above, with $y_{MIN}$ the average of the experimental fluorescence intensity values at 5 and 10 nM $Ca^{2+}$, and $y_{MAX}$ the average of the values at 20 and 25 mM $Ca^{2+}$. Free $Ca^{2+}$ was estimated for each data point from $y$ and the D4 standard curve with $K_d$ 200 μM, as shown conceptually in Fig. 3h. The equation is

$$[Ca^{2+}]_{FREE} = K_d(y/(1-y)), \qquad (3)$$

a rearrangement of Eq. (2). Bound $Ca^{2+}$ was estimated as

$$[Ca^{2+}]_{BOUND} = [Ca^{2+}]_{TOTAL} - [Ca^{2+}]_{FREE}. \qquad (4)$$

Only the data points for total $Ca^{2+}$ 10–500 μM are plotted in Fig. 3i, because there was no measurable binding at lower total $Ca^{2+}$ concentrations, and the

scatter in estimating bound $Ca^{2+}$ by subtracting two large numbers became excessive at higher concentrations.

**Isothermal titration calorimetry.** $Ca^{2+}$ binding to EFSAM variants was measured using a MicroCal ITC-200 microcalorimeter (Malvern) at 20 °C. Chelex-treated buffer containing 50 mM Tris-Cl pH 7.5, 150 mM NaCl, and 5% glycerol was used to dilute the proteins and $Ca^{2+}$ stock. The buffer contained 2 mM $Mg^{2+}$ where indicated. EFSAM-GrpE concentrations were calculated from $A_{280}$ (measured with a NanoDrop spectrophotometer), after subtracting the buffer blank reading, using the tryptophan and tyrosine molar extinction coefficients determined by Edelhoch[38] for tryptophan and tyrosine model compounds. (EFSAM-GrpE contains no cysteine/cystine.) $Ca^{2+}$ was injected 39 times, 1 μl each, with 150 s intervals between injections. The background data obtained from the buffer sample were subtracted before the data analysis. The data were analyzed with the Origin7 software package (MicroCal).

It is possible to read the approximate number $n$ of binding sites from a 'Wiseman plot' of the ITC data, such as the one in the right-hand panel of Fig. 3a. The original paper describing this method of analysis[39] showed that under the experimental conditions used in ITC measurements, and for a protein having a single class of binding sites (that is, all the sites have the same $K_d$ and $\Delta H$), $n$ corresponds approximately to the molar ratio of ligand to protein at the midpoint of the titration curve. This feature carries over when there is more than one class of sites, taking for $n$ the molar ratio at the midpoint of the final phase of the titration curve, as illustrated by a collection of Wiseman plots in the literature[40,41].

Fitting ITC data quantitatively requires adoption of a specific model, such as binding to a single class of sites, binding to two classes of independent sites, or in some cases a more complicated model. We did not attempt to fit the EFSAM ITC data to a detailed binding model because of the large number of $Ca^{2+}$-binding sites, the interdependence of binding at the EF-hand and the other $Ca^{2+}$-binding sites, and the coupling of $Ca^{2+}$ binding to a change in intradimer EFSAM–EFSAM interactions. A quantitative description of binding would entail fitting a large number of $K_d$ values, $\Delta H$ values, parameters for coupling between the sites, and parameters for coupling between binding at individual sites and the dimer > monomer conformational change. Mathematically, the data do not provide adequate constraints to fit so many parameters.

The standard software supplied with the MicroCal ITC-200 microcalorimeter has options to fit data to models with one or two classes of independent sites. We stress that these models are not appropriate for EFSAM, and that any estimated values for $K_d$ and $\Delta H$ are not meaningful. However, the estimate for $n$ may be credible, because any fit has to regress to the baseline in the region around the true value of $n$, independent of the model chosen. Using the MicroCal software to fit the data to a model with two classes of independent sites consistently yielded an estimate of 5–6 binding sites per EFSAM monomer.

**Circular dichroism spectrometry.** CD spectra of EFSAM-GrpE variants were recorded in an Aviv 62 DS spectropolarimeter at ambient temperature (~22 °C), using a 1-mm path length quartz cell with the protein concentration at 5 μM in Chelex-treated buffer containing 50 mM Tris pH 7.5, 100 mM NaCl, 5% glycerol, and 1 mM or 10 mM $Ca^{2+}$ where indicated. All spectra were obtained as the average of at least six scans with a scan rate of 50 nm/min. The ellipticity was measured from 190 nm to 260 nm and converted to mean residue ellipticity (deg $cm^2$ $dmol^{-1}$ $res^{-1}$). For the experiment of Supplementary Fig. 8, CD spectra of EFSAM-GrpE and GrpE proteins were recorded after 2-min equilibration at the specified temperatures between 25 °C and 65 °C.

**Hydrogen-deuterium exchange–mass spectrometry.** The pepsin digestion profile of each EFSAM-GrpE variant was determined using 10 μM protein. For the deuterium exchange, deuterated Chelexed buffer, pD 7.5, was prepared by lyophilization of Chelex-treated buffer containing 50 mM Tris pH 7.5, 100 mM NaCl, 5% glycerol, and resuspension in $D_2O$ (Cambridge Isotopes). In pilot experiments, the number of EFSAM peptides reliably detected by MS after pepsin digestion decreased at $Ca^{2+}$ concentrations higher than 30 μM. This concentration was acceptable for the HDX–MS experiments, since the FRET decrease of donor/acceptor-labelled EFSAM-GrpE was nearly complete at 30 μM $Ca^{2+}$, and hence the defining conformational change induced by $Ca^{2+}$ could be captured at 30 μM $Ca^{2+}$. EFSAM proteins, in the presence or absence of 30 μM $Ca^{2+}$, were diluted to 10 μM in the deuterated Chelexed buffer, quenched after 0.5, 1, 2, and 5 min with 125 mM sodium phosphate monobasic, pH 2.6 for 1 min, and injected into a Waters G2-Si HDX-MS system (Waters Corp.) by a LEAP H/DX PAL autosampler. Non-deuterated protein controls were processed in parallel. Protein samples were digested on-column with immobilized pepsin (Pierce) at 0 °C, then separated by liquid chromatography on a Waters NanoACQUITY UPLC BEH C18 column with a 7–85% acetonitrile gradient in 0.1% formic acid. Peptides were analyzed by electrospray ionization mass spectrometry with ion mobility separation in a Synapt G2-Si quadrupole time-of-flight mass spectrometer. Peptides were identified with ProteinLynx Global Server (Waters Corp.). Mass spectra were assigned and H/D exchange was determined with DynamX 3.0 (Waters Corp.). The data were corrected for a back exchange loss of < 26%.

**Biotinylation of exposed cysteine residues.** HeLa cells ( ~ $6 \times 10^6$ cells per construct) were transfected with the specified single-cysteine constructs using Lipofectamine 2000 and grown under 10% $CO_2$, in DMEM (HyClone media, Thermo) containing 10% heat-inactivated FBS for 24 h. The cells were scraped from the substrate and resuspended in lysis buffer containing 25 mM Tris-Cl (pH 6.8 with NaOH), 25 mM NaCl, 12.5 U/ml DNase I (Thermo Fisher; catalog number 90083), protease inhibitors (Roche; catalog number 11873580001, 1 tablet per 50 ml buffer), 1 mM DTT, with either 0.5 mM EGTA (low $Ca^{2+}$) or 2 mM $Ca^{2+}$ (high $Ca^{2+}$). Cellular membranes were prepared and resuspended in 200 μl resuspension buffer containing 25 mM Tris-Cl (pH 6.8) and 150 mM NaCl, with either 0.5 mM EGTA or 2 mM $Ca^{2+}$. 300 μM biotin-maleimide reagent (Trilink; catalog number B-1012) was added to the resuspended membranes and the reaction was carried out for 1 h at room temperature. The reaction was quenched using 5 mM DTT and 200 mM Tris-Cl (pH 8.5). The membranes were collected by centrifugation at 167,000 g for 20 min in an airfuge (Beckman Coulter) at 4 °C. The pellets were resuspended in 200 μl pulldown buffer containing 50 mM Tris-Cl pH 8.5, and 150 mM NaCl. Once thoroughly resuspended, the membranes were solubilized with 0.5% Triton-X 100, 0.5% NP-40 and 0.5% SDS. The solubilized membranes were diluted further with 1200 μl of pulldown buffer and incubated with MyOne streptavidin T1 Dynabeads (Thermo Fisher). The beads were thoroughly washed thrice with pulldown buffer, boiled in 5x reducing loading/dye buffer to elute the biotinylated proteins and analyzed by SDS-PAGE using 4–12% NuPAGE Bis-Tris gels (Invitrogen). Immunoblotting was performed using rabbit anti-STIM1 (Cell Signaling Technology #4916, 1:1000) and HRP-labelled goat anti-rabbit secondary antibody (Sigma-Aldrich, catalog number A0545, 1:4000) and the blots were developed using ECL substrate (Perkin Elmer).

For assessing the biotinylation of EFSAM-GrpE single-cysteine mutants, the specified proteins were overexpressed in *E. coli* strain BL21 (NEB) and grown at 37 °C. Proteins were purified using Ni-NTA affinity purification as described above, without any added $Ca^{2+}$. The purified proteins were passed through PD Minitrap G-25 desalting columns (GE Healthcare) to remove β-mecaptoethanol and imidazole. The proteins were eluted in biotinylation buffer containing 50 mM Tris-Cl (pH 6.8) and 150 mM NaCl. 24 μM of the specified protein, containing either 0.5 mM EGTA or 2 mM $Ca^{2+}$, and under native conditions (diluted using biotinylation buffer) or under denaturing conditions (diluted using 8 M urea; final urea concentration being 6 M), was treated with an equimolar amount of biotin-maleimide reagent for 5 min at room temperature. The biotinylation reactions were quenched directly with 5x reducing loading/dye buffer and the samples analyzed by SDS-PAGE using 4-12% NuPAGE Bis-Tris gels (Invitrogen). Immunoblotting was performed using HRP-labelled goat anti-biotin antibody (Cell Signaling Technology #7075, 1:6000) and the blots were developed using ECL substrate (Perkin Elmer).

**Cellular STIM1 expression and confocal microscopy.** HeLa cells (ATCC) were transfected with the specified constructs using Lipofectamine 2000 and grown under 10% $CO_2$, in DMEM (HyClone media, Thermo) containing 10% heat-inactivated FBS for 24 h before imaging or crosslinking experiments. HeLa cells, transiently expressing wild-type eGFP-STIM1 or specified variants, were grown on 35-mm glass-bottom dishes (MatTek). Imaging was performed using an Olympus Fv10.1 confocal laser-scanning microscope with a 60×, 1.35 NA, oil-immersion objective. Cells were imaged at room temperature, first in Ringer's buffer containing 20 mM HEPES (pH 7.4 with NaOH), 125 mM NaCl, 5 mM KCl, 1.5 mM $MgCl_2$, 1.5 mM $CaCl_2$, and 10 mM D-glucose, for the resting-state images. This solution was replaced with a modified Ringer's buffer lacking $CaCl_2$ in the presence or absence of 1 μM thapsigargin (Life Tech), as specified, and incubated at least 5 min before obtaining the activated state images.

**CFP-YFP live-cell FRET assay.** HeLa cells co-expressing CFP- and YFP-tagged EFSAM-TC-GrpE protein pairs, corresponding full-length STIM1 protein pairs, or a tandem CFP-YFP construct were grown on 35-mm glass-bottom dishes (MatTek) overnight. The pCMV-CFP-YFP control plasmid was a gift from Liusheng He and Jehonathan Pinthus (Addgene plasmid # 24520). A Zeiss LSM 780 confocal microscope equipped with 405-nm and 514-nm laser lines was used to collect CFP, YFP, and FRET images. The apparent FRET efficiency (expNFRET) between CFP- and YFP-labelled proteins was calculated with the help of PixFRET, an ImageJ-based plugin[42].

**mCherry-CAD relocalization assay.** HeLa cells, co-transfected with mCherry-CAD and the specified variants of STIM2(1-434)-eGFP, were grown overnight in a modified DMEM medium containing 0.2 mM $CaCl_2$ and 10 μM $LaCl_3$ on 35-mm glass-bottom dishes (MatTek). Imaging was performed using an Olympus Fv10.1 confocal laser-scanning microscope with a 60 ×, 1.35 NA, oil-immersion objective. Care was taken to obtain confocal sections at the mid-nuclear level of the cells, showing plasma membrane as well as cytoplasmic compartments. Cells were imaged at room temperature, first in Ringer's buffer with no additional $Ca^{2+}$, followed by Ringer's buffer with 12 mM $Ca^{2+}$. Fluorophore intensities were quantitated in regions of interest at the plasma membrane and cytoplasmic compartments using ImageJ. Statistical significance of the differences was determined using unpaired, two-tailed Student's $t$-test.

For the extracellular $Ca^{2+}$ titration in Supplementary Fig. 6, HeLa cells were plated on 35-mm glass bottom cell culture dishes and transfected with STIM2(1-434)-eCFP and mCherry-CAD plasmids in DMEM medium. 18 h post-transfection, DMEM medium was switched to an imaging solution comprising 150 mM NaCl, 7.2 mM KCl, 1.2 mM $MgCl_2$, 11.5 mM glucose, 20 mM HEPES adjusted to pH 7.2 with NaOH. After incubation in the $Ca^{2+}$-free imaging solution for 10 min, live-cell imaging was carried out on a Nikon Eclipse Ti-E microscope (Nikon Instruments) equipped with an A1R-A1 confocal module with LU-N4 laser sources (argon-ion: 405 and 488 nm; diode: 561 nm), CFI (chrome-free infinity) plan Apochromat VC series objective lenses (40 × oil), and a live-cell culture cage to maintain the cells at 37 °C and 5% $CO_2$. Time-lapse imaging was performed to monitor the subcellular localization of mCherry-CAD upon addition of increasing amounts of external $Ca^{2+}$ (0–25 mM). The cytosolic intensity of mCherry-CAD was analyzed by using the NIS-element AR software (version 4.0).

**STIM1 $Ca^{2+}$ concentration-dependent crosslinking assay.** STIM1 crosslinking assay was performed as described in ref. [28]. Briefly, HeLa cells expressing specified variants of STIM1(A230C) (~$12 × 10^6$ cells) were scraped from the substrate and resuspended in Chelex-treated (Chelex-100 resin, Bio-Rad) dilution buffer with no added EGTA or $Ca^{2+}$. Cellular membranes were prepared and resuspended in 130 μl Chelex-treated resuspension buffer containing 25 mM Tris-Cl, pH 7.5, 150 mM NaCl, and 0.3 mM DTT. Equal volumes of membranes were apportioned to twelve wells containing resuspension buffer supplemented to give either a final EGTA concentration of 0.5 mM or final $Ca^{2+}$ concentrations ranging from 0.3 μM to 2 mM. Iodine oxidation, SDS-PAGE analysis, immunoblotting with anti-STIM1 antibody (Cell Signaling Technology #4916, 1:1000), and quantitation were as described in ref [28].

**Single-cell $Ca^{2+}$ influx assay.** Single-cell $Ca^{2+}$ imaging was performed using HeLa cells that had been co-transfected with eGFP-STIM1 variants and mCherry-ORAI1 (ref. [43]), FACS-sorted for medium-level GFP-expressing cells, and plated on 18-mm coverslips. The cells were loaded with 5 μM Fura-2-acetoxymethyl ester for 45–60 min at 37 °C in DMEM containing 0.02% Pluronic F-127 and 10 mM HEPES pH 7.4, washed twice with fresh media, and analysed immediately. Modified Ringer's solution used in this assay consisted of 125 mM NaCl, 5 mM KCl, 1.5 mM $MgCl_2$, 10 mM D-glucose, and 20 mM HEPES (pH 7.4 with NaOH), with the addition of either 1 mM or 2 mM $CaCl_2$, or 1 μM TG, where indicated. Coverslips were assembled into a chamber on the stage of an Olympus IX 71 microscope equipped with an Olympus UPLSAPO 20 × , NA 0.75, objective. Cells were alternately illuminated at 340 nm and 380 nm with the Polychrome V monochromator (TILL Photonics) using ET - Fura2 filter (Chroma Technology Corp., catalog number 79001). The fluorescence emission at $\lambda > 400$ nm (LP 400 nm, Emitter 510/80 nm) were captured with a CCD camera (SensiCam, TILL Imago). Ratio images were recorded at intervals of 4 s. $Ca^{2+}$ concentration was estimated from the relation[44]

$$[Ca^{2+}]_i = K_d((R - R_{min})/(R_{max} - R))\left(S_{f2}/S_{b2}\right), \qquad (5)$$

where $K_d = 220$ nM, and the values of $R_{min}$, $R_{max}$, and $(S_{f2}/S_{b2})$ were determined from an in situ calibration of Fura-2 in HeLa cells. Data were analyzed using TILL Vision (TILL Photonics).

**Experimental replicates and statistical analysis.** Data in Figs. 5a and 5c are representative of five experiments. Data in Figs. 1d, 3, 7a, b, d, and e are representative of three experiments. Data in Figs. 1f, g, 3a–c, e–g, 5b, and 6 are representative of two experiments. The conclusions from Fig. 6 for GFP-STIM1 cysteine mutants are supported by data from two similar experiments with HA-STIM1 cysteine mutants, which are not included here. Statistical significance of observed differences was determined using an unpaired, two-tailed Student's t-test.

## Data availability
Data supporting the findings of this manuscript are available from the corresponding author upon reasonable request. HDX–MS data have been deposited in the MassIVE database maintained by UCSD Centre for Computational Mass Spectrometry under accession code MSV000082950.

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

## Acknowledgements

We thank Dr Anjana Rao for discussions and comments on the manuscript. This work was funded by NIH grants AI084167 and GM110397 (to P.G.H.), AI040127 (to Anjana Rao and P.G.H.), NIH Shared and High-End Instrumentation Award S10OD21724 (to E.A.K.), NIH grants GM112003 and GM126532 (to Y.Z.), American Cancer Society grant RSG-16-215-01 TBE (to Y.Z.), the National Natural Science Foundation of China grant NSFC-31471279 (to Y.W.), Fundamental Research Funds for the Central Universities 2017EYT21 (to Y.W.), and postdoctoral fellowship 2016/12505-8 from the São Paulo Research Foundation (FAPESP) (to A.E.Z.).

## Author contributions

A.G. and P.G.H. designed the experiments. A.G. was the lead scientist for all the protein engineering, biochemical and biophysical measurements, and cellular assays. A.E.Z. joined in identifying and characterizing STIM1-2NQ(6-11). N.H. contributed to the molecular cloning, protein purification, crosslinking, and fluorescence assays. V.R. developed the STIM1 crosslinking assay. A.A.B. carried out the ITC measurements. G. M., S.Z., Y.W., and Y.Z. engineered plasma membrane-anchored STIM, and developed the mCherry-CAD relocalization assay and advised on its application to the STIM2-2NQ mutant. E.A.K. guided and participated actively in the HDX–MS experiments. A.G. and P.H. analyzed the data and wrote the manuscript with input from the other authors.

## Additional information

**Competing interests:** P.G.H. is a founder and scientific advisor of CalciMedica, Inc. The other authors declare no competing interests.

