## [Peer Review File · Nature Communications]

Reviewers' Comments:

Reviewer #1:

Remarks to the Author:

The manuscript "Calcium sensing by the STIM1 ER-luminal domain" aims at re-evaluating the mechanism of STIM1 activation through dissociation of Ca²⁺ from a single EF hand site leading to structural re-arrangements and dimerization of the STIM1 luminal domain. In contrary, STIM1 luminal domain apparently has 5-6 undetected Ca²⁺ binding sites, which are energetically coupled to the EF hand site. Moreover, Ca²⁺ dissociation controls a second structured conformation of the luminal domain. The additional luminal domain Ca²⁺ binding sites interact with the EF hand to control physiological STIM1 activation.

The manuscript is interesting and the numerous methods employed are impressive. Nonetheless, there a number of questions that need to be addressed. Additionally, the manuscript is difficult to read, particularly for the broad readership of this Journal.

1) One of the main questions is, whether the linkage of the EFSAM domain with GrpE may indeed replicate characteristics of wild-type STIM1. GrpE may also influence features of the the EF-SAM domain in a different manner than STIM1, particularly with respect to the distinct Ca²⁺ binding affinities. In this context, the time-courses from FRET experiments with wild-type CFP/YFP-STIM1 and CFP/YFP-EFSAM-TC-GrpE following store depletion should be shown. What will be the difference when in the latter construct the CC1 domain is omitted?

2) The new aspect of the initiation event from this manuscript is the notion that there are 5-6 Ca²⁺ binding sites on EFSAM. However, none of Ca²⁺ binding experiments are performed in the presence of physiological levels of Mg²⁺ (i.e. ~1-2 mM Mg²⁺). Also, none of the Ca²⁺ binding experiments (or any biochemical assessments - CD, HD-X, ANS, competition binding, FRET) are performed at near physiological temperature which will dramatically influence the weak interactions being detected.

3) Why was only 30 μ M Ca²⁺ used in the HD-X experiments? An entirely different set of HD-X protection data will be observed at physiological Ca²⁺ levels and temperature.

4) The authors claim that there are 5-6 Ca²⁺ binding sites, with a single point mutation of the EF-hand (D76A) eliminating all apparent Ca²⁺ binding. Further, the authors claim that a Ca²⁺ binding site is occupied with ~10 microM affinity. A direct 45-Ca²⁺ binding assay (in the presence of physiological Mg²⁺ levels) would easily detect binding sites in this affinity range.

5) The notion that the luminal domain remains largely structured after losing Ca²⁺ and dimerizes (rather than oligomerizes) is not new. The dimer has been identified as a Ca²⁺ depleted conformation in 2006 (PMID 17020874), and it has been shown in 2007 as well as 2009 that a large loss in structure is not a requirement for dimerization and activation of STIMs (PMID 19019825, 18166150). Further, isolated (Ca²⁺-depleted) EFSAM, in fact, retains partial structure. Additionally, the current authors themselves have previously shown (PMID 23851458, 26184105) that luminal domain dimerization is the key initiation event.

6) The thermodynamic model shown in Fig. 7a is problematic. First, no quantitative thermodynamic data have been reported in this study. Second, the data show that elimination of the EF-hand Ca²⁺ binding site via D76A abolishes all apparent Ca²⁺ binding sites, so the lower pathway in Fig. 7a has not even been demonstrated to exist (which is additionally stated in Fig. 7b). Third, there is no Ca²⁺ shown bound to EFSAM in the low Ca²⁺ concentration (i.e. less than 100 microM) even though the authors propose there is a 10 μ M Ca²⁺ binding site (Fig. 7b)

Additional points:

Please provide a more detailed explanation of GrpE. Is it structurally related to STIM1/ STIM1 C-term? What is known of its dimer affinity? Are there mutations known to increase or decrease dimerization in an attempt to see whether this may affect the Ca²⁺ binding affinities at the EFSAM domain.

Fig. 1g - Donor fluorescence should be systematically increasing as acceptor fluorescence decreases. This trend is not observed?

Typically, protein concentration is determined on samples used in ITC experiments using amino acid analysis for accurate stoichiometric determinations. There is no mention of how protein was quantified in any of the experiments reporting stoichiometry information. This type of accuracy is particularly important given that stoichiometry is the only parameter the ITC is being used to extract.

The three NQ mutations should be described in the Results section together with the rationale to improve readability.

How is the structure altered with EFSAM-6-11 in comparison to 2NQ?

What is the rationale behind all peptides in Fig. 5 – please explain in more detail.

It would be interesting to compare constitutive Ca²⁺ entry of D76A with EFSAM-6-11.

Fig.3 e-g: color codes are mixed.

Reviewer #2:

Remarks to the Author:

Gudlur et al report a significant advance in calcium sensing by the STIM proteins, which are ubiquitous ER Ca²⁺ sensors in most animal cells. How STIM1 senses ER Ca levels and transduces Ca concentration changes into store-operated Ca entry is a topic of great intense but remains incompletely understood. Rationalizing that the luminal Ca²⁺ sensing region of STIM1 (EF-SAM) might more faithfully represent the behavior of native protein when it is stably dimerized (mimicking the constitutive dimeric state of STIM1), the group begins their analysis by examining the EF-SAM domain tethered to a soluble dimerizing GrpE domain from *Thermus thermophilus*. Analysis of Ca²⁺-dependent FRET of the EF-SAM GrpE construct yields the unexpected result that the GrpE fusion proteins of EFSAM display a substantively heightened Ca²⁺ binding sensitivity with a K_d below 10 μM, compared to isolated EF-SAM and STIM1 proteins which were previously reported to respond with K_d values in the 100s of μM. Critically, ITC and an innovatively designed D4-sensor based fluorescence studies show the presence of multiple binding sites, which are traced to potentially arise from acidic EFSAM surface residues. The study further finds that contrary to current views derived from the isolated EFSAM domains, Ca²⁺ unbinding does not lead to unfolding of its secondary structure, which forces a revision of the prevailing views of Ca²⁺ sensing mechanism of STIM proteins. Instead, it is proposed that Ca²⁺-unbinding leads to a switch in the secondary structure of EFSAM, presumably putting into motion the release of the cytosolic SOAR/CAD domain from CC1. Overall, this paper provides a fresh, thought-provoking perspective on Ca²⁺ binding in the EFSAM domain and STIM1 activation. The experiments are well designed and most conclusions are well supported by the results. However, the paper needs to address some key questions and potential problems as described below. The overall conclusion that EFSAM does not become disordered during Ca²⁺ unbinding is convincing, but the paper could be further improved if some of the surface Ca²⁺ binding sites could be identified and characterized.

-The first half of the paper relies heavily on data generated in the EFSAM-GrpE construct and therefore it is very important to demonstrate that this fusion protein provides a reliable readout for how EFSAM domain behaves in the context of STIM1. How does the dimerization propensity of GrpE compare with that of STIM1? The authors state that the EFSAM domain behaves differently (i.e. does not unfold as easily in the absence of Ca²⁺) when linked to GrpE. While stabilizing it with GrpE is useful from a technical perspective, could this obscure some conformational changes

that occur in STIM1? This is a key point that could derail the conclusions drawn and needs to be further validation beyond the minimal set of controls shown in Figure 1.

-The conclusion that the higher Ca²⁺ concentration dependence of full-length STIM1 relative to the EF-SAM-GrpE construct used here could be due to an "energetic cost exacted by the coupling between Ca²⁺ binding" and ensuing conformational changes is a vague and unsatisfactory explanation for the widely differing Ca²⁺ sensitivities of EF-SAM and STIM1. This critical divergence needs to be more adequately explained. Could the presence of the TM domain between CC1 and the EFSAM in full-length STIM1 offer constraints that lower the Ca²⁺ sensitivity of EFSAM in its native environment? If so, are the conclusions drawn here without the TM domain extendable to full-length STIM1?

-Figure 2a,b,d. EFSAM-TM-CC1-GrpE construct was used to parse out the relative contribution of EFSAM domain vs. SOAR/CAD domains to oligomerization observed via FRET. The conclusion that intradimer rearrangement of EFSAM domains accounts for most of the STIM1-STIM1 FRET seen with full-length STIM1 is entirely based on the levels of similar FRET seen between the two constructs....this could be purely correlative and seems like significant over-reach based on simple experiment. More evidence is needed to bolster this strong claim.

-Related to the above point, the conclusion that "intradimer rearrangement of EFSAM domains accounts for an appreciable part of the STIM1-STIM1 FRET change" – the data do not rule out that part of the FRET change could arise from oligomerization of CC1 domains downstream of Ca²⁺ unbinding from the EFSAM domains?

-Tamas Balla's group likewise addressed a closely related question in previous work (PMID: 28724757). This study needs to be cited and discussed in this context.

-An important point is to determine how do STIM1-1NQ and STIM1-3NQ mutants affect Ca²⁺ binding to the EFSAM domain? Resolution of this question could be important for the key tenets of this study that Ca binding to the EF hand domain could be stabilized by interaction of EF hand with the SAM domain.

-What is the main difference between STIM1-2NQ and STIM1-2NQ (6-11) that switches the protein from a loss-of-function phenotype into a gain-of-function one?

-Figure 5. While the supposition that substantive loss of secondary structure of EFSAM-GrpE should be detectable by CD based on the argument that 50% of the helical content is from EFSAM, this experiment needs a positive control. A comparison within the same experiment to the behavior of EFSAM would be in order here.

-Based on supplementary Fig. 6a, does STIM2 EFSAM domain have more Ca²⁺ binding sites (15?) than STIM1?

Minor comments:

-The text needs to provide greater detail regarding exactly how multiple Ca binding sites are derived from the ITC and the D4 sensor data...it is not be entirely clear to the general reader how this follows from the results shown. Related, the full equations used and the parameters of the fit for Figure 3a, e-h and especially the conclusions from panel "i" need to be more explicitly spelt out.

-Related, what happens when the data are fit by fewer than 5 sites? Fits to 1, 2, 3, 4, 5 or more binding sites could be shown to demonstrate the inadequacy of the single or double binding sites.

-First sentence of "Ca²⁺-dependent conformational change in cells" section is a little ambiguous. It would be helpful to specify that they are referring to the increase in STIM1-STIM1 FRET during

store-depletion.

-Fig. 3 e-g: Labels and markers don't match

-Sentence describing Fig. 6e on page 14 needs explanation of how the A230C assay works or alternatively, the result can be moved to supplementary information.

Reviewer #3:

Remarks to the Author:

STIM1 molecule is essential in store-operated calcium entry process. The previous fundamental view is that STIM1 senses calcium depletion inside endoplasmic reticulum and then aggregates together to become activated, eventually leading to calcium influx through Orai1 channel on plasma membrane. This manuscript by Gudlur et al proposes a novel viewpoint that multiple calcium binding sites interacting with the EF-hand site of EF-SAM domain somehow results in the activation of STIM1 in cells. The major evidence comes from the results that authors designed a new soluble STIM1 construct in which EF-SAM domain coupled through a short linker to GrpE from *Thermus thermophilus*. This new EFSAM-GrpE is soluble and did not adopt drastic conformational changes with or without calcium binding. Interestingly, this soluble protein has 5-6 calcium binding sites. Overall, this paper provided enough evidence to support their viewpoint. And this viewpoint will substantially revise current understanding of calcium sensing by STIM1. However, several critical questions need to be answered before the acceptance of this manuscript.

1. The manuscript nicely demonstrates that calcium binding induces the subtle conformational changes of EF-SAM domain. However, this conclusion is based on artificial construct EFSAM-GrpE. This result must be confirmed in the full-length STIM1 molecule wild type or near native STIM1 construct (23-444) *in vitro*. This is very important because different STIM1 constructs or fusion proteins will have different behaviors. Physiologically, full-length STIM1 wild type is the one to carry out function inside cell.

2. The authors used ITC and fluorescence assay to show that calcium binds to multiple sites of EF-SAM domain. EFSAM-GrpE is a dimer and contains at least two calcium binding EF-hands. It is easy to understand that two EF-hands may have cooperative binding. This explains the findings from Figure 3a. Thus, multiple calcium binding sites conclusion is not suitable for ITC experiment. The next important question is that where are those binding sites? Which specific residues they are.

3. It is puzzling that STIM1-2NQ mutant (D77N, D82N, E86Q, D89N, E90Q, E94Q, D100N, E111Q, D112N, E118Q, and D119N) is inactive from the Figure 4b and contains one calcium binding site from the Figure 4c, however, STIM1-2NQ(6-11) (E94, D100, E111, D112, E118, and D119) is self-active from Figure 6a,c and contains multiple calcium binding sites from Figure 6d. These results seem to be contradictory. The authors might address the reason behind these inconsistent results.

4. back to fundamental question. If the EF-SAM does not adopt drastic conformational changes upon calcium loss, what force drives the conformational changes of CC1 region to expose SOAR or CAD domain. Where does the energy come from? How to explain the characteristic feature of STIM1 punta upon activation ?

Gudlur *et al* NCOMMS-18-09640-T / Responses to reviewers / 16aug18

We thank all the reviewers for their extensive and constructive criticisms. Responses to their comments are given below. Notable additions to the revised manuscript, prompted by the reviewers' suggestions, are that binding to the EFSAM sites is Ca²⁺ specific, and that EFSAM in full-length STIM1 does not unfold upon Ca²⁺ depletion.

Reviewer #1 (Remarks to the Author):

The manuscript "Calcium sensing by the STIM1 ER-luminal domain" aims at re-evaluating the mechanism of STIM1 activation through dissociation of Ca²⁺ from a single EF hand site leading to structural re-arrangements and dimerization of the STIM1 luminal domain. In contrary, STIM1 luminal domain apparently has 5-6 undetected Ca²⁺ binding sites, which are energetically coupled to the EF hand site. Moreover, Ca²⁺ dissociation controls a second structured conformation of the luminal domain. The additional luminal domain Ca²⁺ binding sites interact with the EF hand to control physiological STIM1 activation.

The manuscript is interesting and the numerous methods employed are impressive. Nonetheless, there a number of questions that need to be addressed. Additionally, the manuscript is difficult to read, particularly for the broad readership of this Journal.

1) One of the main questions is, whether the linkage of the EFSAM domain with GrpE may indeed replicate characteristics of wild-type STIM1. GrpE may also influence features of the the EF-SAM domain in a different manner than STIM1, particularly with respect to the distinct Ca²⁺ binding affinities. In this context, the time-courses from FRET experiments with wild-type CFP/YFP-STIM1 and CFP/YFP-EFSAM-TC-GrpE following store depletion should be shown. What will be the difference when in the latter construct the CC1 domain is omitted?

The question has two levels: broadly, to what extent does EFSAM linked to GrpE afford insights into the functioning of full-length STIM1; and, more specifically, what can be learned from further study of EFSAM-TC-GrpE. We address the narrower question first.

The purpose of the EFSAM-TC-GrpE experiment was to determine the extent of EFSAM–EFSAM FRET in cells under conditions where there could be no contribution from SOAR/CAD-dependent oligomerization. It is clear that there was a substantial FRET change upon store depletion, so the original question has been answered. Could we learn more from additional experiments on EFSAM-TC-GrpE and its engineered variants? Certainly. We could explore the role of CC1-CC1 dimerization by replacing the CC1 regions with nondimerizing single α -helices, or investigate possible constraints on EFSAM due to EFSAM interactions with the ER membrane. But these issues lead in new directions, and are beyond the scope of this manuscript.

The broader question regarding the effect of replacing STIM1(210–685) with GrpE has also been raised by the other reviewers. Conceptually, the effect might lead in either of two directions. Replacing STIM1(210-685) might relieve constraints ordinarily placed on EFSAM by its protein and cellular context— such as those imposed by the TM helices, by tethering to the ER membrane, and by the conformational change in CC1-SOAR— or it might create new constraints that did not exist in the native STIM1 protein. We have discussed specific cases in the manuscript where the STIM1 EF-hand has been assayed in different protein contexts (EF-hand grafted into a domain from CD2 *versus* EFSAM-GrpE; isolated EFSAM domain *versus* EFSAM-GrpE; EFSAM-GrpE *versus* full-length STIM1) and the likely reasons for the corresponding shifts in Ca²⁺ binding.

Indeed, the dependence of Ca²⁺ binding on the protein and cellular context is not limited to comparisons between full-length STIM1 in ER membranes and the engineered construct EFSAM-GrpE *in vitro*. The

Zheng *et al* manuscript that we provided along with our submission documents a clear effect of the cellular context on the Ca²⁺ affinity of wildtype full-length STIM proteins— the midpoints of a Ca²⁺ titration for STIM proteins expressed at the plasma membrane (0.97 μM for STIM1, 1.48 μM for STIM2) differ from the values measured for the same proteins in the ER.

The EFSAM-GrpE constructs themselves offer another illustration. We have seen a consistently lower Ca²⁺ midpoint (~1 μM) in titrations of the single α-helix linker (SAH) construct of Supplementary Figure 1d-h than in titrations of the EFSAM-GrpE construct (~10 μM) used in Figure 1e-g and in all other experiments. The probable explanation is that the increased set of configurations available to EFSAM and the single α-helix linker after dissociation of the EFSAM-EFSAM dimer translates to increased entropy, and hence a less unfavorable EFSAM dimer > EFSAM monomer free energy change. Consequently a lower concentration of Ca²⁺ suffices to drive dissociation of the dimer.

These examples led us to the important conclusion that the interdependence of Ca²⁺ binding and protein conformation can determine the effective Ca²⁺ binding affinity of the STIM luminal domain.

The next logical question is, if there are changes in EF-hand Ca²⁺ binding, then what characteristics of STIM1 are preserved in EFSAM-GrpE? First, since isolated EFSAM is well-folded and stable in its Ca²⁺-bound form, it can be expected that EFSAM assumes this same structure in full-length STIM1 and in EFSAM-GrpE when the EF-hand site is occupied. Second, having the same folded structure implies that these proteins present the same EFSAM surface Ca²⁺-binding sites. The possibility that close linkage of the EFSAM domain to GrpE produced the experimentally observed sites artifactually is dispelled by the finding that ITC still reflects ~5 Ca²⁺-binding sites in EFSAM-SAH-GrpE, where EFSAM is decoupled physically from GrpE by the long SAH spacer. Third, we have shown that the luminal domain of full-length STIM1 retains its secondary structure in Ca²⁺-free conditions, in an experiment described in our response to Reviewer #3, comment 1 (see new Figure 6 and new Supplementary Figure 11).

2) The new aspect of the initiation event from this manuscript is the notion that there are 5-6 Ca²⁺ binding sites on EFSAM. However, none of Ca²⁺ binding experiments are performed in the presence of physiological levels of Mg²⁺ (i.e. ~1-2 mM Mg²⁺). Also, none of the Ca²⁺ binding experiments (or any biochemical assessments - CD, HD-X, ANS, competition binding, FRET) are performed at near physiological temperature which will dramatically influence the weak interactions being detected.

We thank the reviewer for suggesting an important additional experiment, measurement of Ca²⁺ binding to EFSAM-GrpE in the presence of physiological levels of Mg²⁺. ITC measurements in the presence of 2 mM Mg²⁺ indicate that there are 4–6 Ca²⁺-specific sites per EFSAM monomer. We have included the data in the revised manuscript as new Supplementary Fig. 2.

We recognize, as well, that the rates of biochemical and cell-biological processes are sensitive to temperature. The concern is mitigated by the fact that STIM1 Ca²⁺ sensing is not markedly influenced by temperature, in our experience, as we see no overall difference in STIM1 relocalization in cells responding to store depletion at room temperature and at 37C. However, the deciding consideration for our experimental conditions was that the key experiments defining STIM1 function in cells, including STIM–STIM FRET upon Ca²⁺ store depletion (Liou *et al* 2007) and the Ca²⁺ concentration dependence of STIM activation (Brandman *et al* 2007; Luik *et al* 2008), were conducted at room temperature. We have carried out our Ca²⁺ binding experiments under comparable conditions. The rationale for this choice of conditions is explicitly noted in the revised manuscript.

3) Why was only 30 μM Ca²⁺ used in the HD-X experiments? An entirely different set of HD-X protection data will be observed at physiological Ca²⁺ levels and temperature.

This concentration of Ca²⁺ was chosen for a straightforward technical reason: At higher Ca²⁺ concentrations, the number of EFSAM peptides reliably detected by MS after pepsin digestion decreases. (This explanation has been included in the revised Online Methods.) The fact that there is already a major change in protein backbone accessibility at 30 μM Ca²⁺ compared to 0 Ca²⁺ shows that 30 μM Ca²⁺ evokes a conformational change in EFSAM. Two further observations strengthen the argument. The FRET decrease of donor/acceptor-labelled EFSAM-GrpE is nearly complete at 30 μM Ca²⁺, so the defining conformational change induced by Ca²⁺ is captured in the HDX-MS experiments at 30 μM Ca²⁺. And the HDX-MS deuterium labelling of wildtype EFSAM at 30 μM Ca²⁺ mirrors that of EFSAM-2NQ, a variant that we confirmed is inactive in Ca²⁺ sensing, establishing that wildtype EFSAM has transitioned to an inactive form at this Ca²⁺ concentration.

Like the reviewer, we believe that there may be a further conformational change at higher levels of Ca²⁺. We noted on p. 13 of original version of the manuscript that ‘The D4 Ca²⁺ sensor titration experiments (Fig. 3g,i) and ANS fluorescence measurements (Supplementary Fig. 8d) indicate that further Ca²⁺ binding to wildtype EFSAM occurs at Ca²⁺ concentrations up to 300–400 μM, and this may correspond to a further consolidation of the inactive conformation.’

That said, it would be a distraction to focus on what happens at higher Ca²⁺ concentrations—the well documented conclusion that EFSAM is structured when Ca²⁺ is bound is not at issue. The main point of our far-UV CD and HDX-MS experiments is that EFSAM is also structured in the absence of Ca²⁺.

4) The authors claim that there are 5-6 Ca²⁺ binding sites, with a single point mutation of the EF-hand (D76A) eliminating all apparent Ca²⁺ binding. Further, the authors claim that a Ca²⁺ binding site is occupied with ~10 microM affinity. A direct 45-Ca²⁺ binding assay (in the presence of physiological Mg²⁺ levels) would easily detect binding sites in this affinity range.

Agreed that ⁴⁵Ca binding is an alternative assay. But the D4 experiment is not fundamentally different from the ⁴⁵Ca assay—it just determines bound and free Ca²⁺ using a different experimental readout.

With regard to the 10 μM site: The D4 assay is tuned to sites binding Ca²⁺ at ~100 μM, as noted. The tuning was intentional, because the point of the assay was to document the presence of sites binding at ~100 μM and to estimate their number. It would be possible to retune the fluorescence assay to lower concentrations of Ca²⁺ using a different Ca²⁺ sensor, but further probing of this concentration range would be redundant, since the FRET, ANS, and HDX-MS experiments all show that there is Ca²⁺ binding in the ~10 μM concentration range.

5) The notion that the luminal domain remains largely structured after losing Ca²⁺ and dimerizes (rather than oligomerizes) is not new. The dimer has been identified as a Ca²⁺ depleted conformation in 2006 (PMID 17020874), and it has been shown in 2007 as well as 2009 that a large loss in structure is not a requirement for dimerization and activation of STIMs (PMID 19019825, 18166150). Further, isolated (Ca²⁺-depleted) EFSAM, in fact, retains partial structure. Additionally, the current authors themselves have previously shown (PMID 23851458, 26184105) that luminal domain dimerization is the key initiation event.

The existence of a luminal domain dimer is a starting point for the manuscript, not a new conclusion from this work. We did cite the earliest report of an EFSAM-EFSAM dimer (Stathopoulos *et al* 2006), and we have added references to Zheng *et al* 2008 and Stathopoulos *et al* 2009 to consolidate the point. That STIM1 EFSAM ‘remains largely structured’ in the absence of Ca²⁺ was not the interpretation of Stathopoulos *et al* 2008 (61% α-helix decreased to 15% in the absence of Ca²⁺) or of Furukawa *et al* 2014 (>50% α-helix decreased to ~10%), nor is it the current consensus view in the field, as noted by Reviewers #2 and #3.

6) *The thermodynamic model shown in Fig. 7a is problematic. First, no quantitative thermodynamic data have been reported in this study. Second, the data show that elimination of the EF-hand Ca²⁺ binding site via D76A abolishes all apparent Ca²⁺ binding sites, so the lower pathway in Fig. 7a has not even been demonstrated to exist (which is additionally stated in Fig. 7b). Third, there is no Ca²⁺ shown bound to EFSAM in the low Ca²⁺ concentration (i.e. less than 100 μM) even though the authors propose there is a 10 μM Ca²⁺ binding site (Fig. 7b).*

There are two different conceptual frameworks for thinking about the results on Ca²⁺ binding. We prefer to think of Ca²⁺ binding in terms of the thermodynamic model, which holds true even if the probability of observing the lower pathway experimentally is negligible. In practice, we believe that the EFSAM surface sites are either masked in EFSAM(D76A) by dimerization, or rearranged in the dimer to a configuration where Ca²⁺ binds with a lower affinity beyond the range detectable in the ITC experiment. The thermodynamic analysis is that, however favorable the free energy of dimer formation, $\Delta G_{\text{dimerization}}$ is not infinite. Dimeric EFSAM will be in equilibrium with some fraction of monomeric EFSAM that can bind Ca²⁺, and therefore it is proper to include the lower pathway and its $\Delta G_2'$ and $\Delta G_1'$ values in the analysis.

Some readers may find a non-thermodynamic approach more intuitive. For example, in the extreme case envisioned by the reviewer, where the lower pathway does not exist, the EF-hand site cannot release Ca²⁺ until Ca²⁺ has dissociated from the other sites. Here Ca²⁺ bound at the surface sites serves as a latch. The net effect is similar—the control of EFSAM conformation by Ca²⁺ dissociation from the EF-hand is intrinsically coupled to Ca²⁺ dissociation from the EFSAM surface sites. The extension of this line of thought to the case where the transit along the lower pathway is possible, but rare, is straightforward.

Figure 7b (Figure 8b in the revision) portrays a model in which Ca²⁺ binding at the EF-hand correlates with and controls the conformational change of full-length STIM1, as stated in the figure legend. This scenario—that the EF-hand controls the conformational change—implies that half-maximal dissociation of Ca²⁺ from the EF-hand must occur at ~200 μM, the value determined from measurements of STIM1 activation in cells. There is no 10 μM site. The Discussion takes note of an alternative model in which the EF-hand site is a structural Ca²⁺-binding site from which Ca²⁺ does not dissociate under physiological conditions. The latter case seemed so straightforward to visualize that we did not include it as an additional figure panel, but we are willing to do so if requested by the editors or reviewers.

Additional points:

Please provide a more detailed explanation of GrpE. Is it structurally related to STIM1/ STIM1 C-term? What is known of its dimer affinity? Are there mutations known to increase or decrease dimerization in an attempt to see whether this may affect the Ca²⁺ binding affinities at the EFSAM domain.

GrpE is not structurally related to STIM1 except for the presence of extended α -helices that form a coiled coil. (The coiled coil is constitutive in GrpE, unlike in STIM1.) This information has been added to the Figure 1 legend. The dimer affinity of the natively folded protein has not been measured, to our knowledge, nor have mutations that compromise dimerization been reported. However, since there is no reason to believe that STIM is regulated by a dimer–monomer equilibrium of SOAR/CAD domains, in our view, GrpE dimer mutants would not add to the understanding of STIM1.

Fig. 1g - Donor fluorescence should be systematically increasing as acceptor fluorescence decreases. This trend is not observed?

The reviewer is correct that an increase in donor fluorescence is expected. We believe the increase is masked by the interfering background signal at wavelengths across the donor peak (see Figure 1f, Acceptor alone; Supplementary Figure 1f, Acceptor alone). We have established that the source of this background signal is the Chelex-based decalcification procedure, but we have not been able to determine the identity of the interfering material(s). The interfering signal is variable from experiment to

experiment, and fluctuates with time. The FRET change is relatively small in any case, so small increases or decreases in the interfering background signal are capable of masking changes in the donor peak intensity due to altered donor-acceptor FRET.

There is at least a hint that donor fluorescence increases. At 570 nm, where this fluctuating signal makes a negligible contribution, there is an increase in the donor fluorescence in the relevant Ca^{2+} concentration range (0–100 μM in Figure 1g; 0–10 μM in Supplementary Figure 1g), which correlates with the decrease in the acceptor fluorescence. The increase in the donor signal at 570 nm is modest, in keeping with the modest change in the acceptor signal.

Typically, protein concentration is determined on samples used in ITC experiments using amino acid analysis for accurate stoichiometric determinations. There is no mention of how protein was quantified in any of the experiments reporting stoichiometry information. This type of accuracy is particularly important given that stoichiometry is the only parameter the ITC is being used to extract.

We agree with the reviewer that accurate determination of the protein concentration is needed in the ITC experiments. We calculated EFSAM-GrpE concentration from A_{280} (measured with a NanoDrop spectrophotometer), after subtracting the buffer blank reading, using the tryptophan and tyrosine molar extinction coefficients determined by Edelhoch for tryptophan and tyrosine model compounds (1967; Biochemistry 6, 1948–1954). (EFSAM-GrpE contains no cysteine/cystine.) A statement to this effect has been added to Online Methods. The NanoDrop has a stated accuracy within 2% (Thermo Scientific), and Gill and von Hippel (1989; Analytical Biochemistry 182, 319–326) showed that molar extinction coefficients of proteins can be calculated with standard deviation $\pm 5\%$ using the Edelhoch coefficients.

The main limitation of this method is that the single-wavelength measurement with the NanoDrop precludes a correction for light scattering based on additional absorption measurements at 320 nm and 360 nm. However, we have no reason to think the light-scattering correction would be appreciable in our samples, and, importantly, any uncorrected signal from light scattering would result in an overestimate of EFSAM-GrpE concentration, and an underestimate of the number of Ca^{2+} binding sites per monomer.

The three NQ mutations should be described in the Results section together with the rationale to improve readability.

We have moved the description and rationale of the NQ constructs from Supplementary Text to Results, as suggested.

How is the structure alter-ed with EFSAM-6-11 in comparison to 2NQ?

We have tested whether there are structural alterations in EFSAM-2NQ(6–11) using the most direct measures available, CD spectroscopy to assess secondary structure, and ANS fluorescence to assess the response to Ca^{2+} [FIGURE R1 for reviewers]. The CD spectra were similar to those of wildtype EFSAM-GrpE, and exhibited a similar response to Ca^{2+} . ANS fluorescence underwent the same reduction with addition of Ca^{2+} as for the wildtype and 2NQ(6-11) proteins, and showed an intermediate blue shift at high concentrations of Ca^{2+} . It is tempting to speculate that the intermediate blue shift corresponds to a less complete stabilization of the inactive conformation in high Ca^{2+} , but this is only one possible interpretation of the data, and we are not ready to draw specific conclusions from the limited data.

What is the rationale behind all peptides in Fig. 5 – please explain in more detail.

Figure 5 reports data for each peptide that could be unambiguously identified by MS in the digested samples. We first optimized the conditions for peptide identification, then acquired deuterium exchange data for all peptides that passed set criteria for ion counts, retention time, and m/z error. We were aided by co-author Dr E Komives' extensive experience with this technique, and we have obtained excellent coverage of the EFSAM sequence.

It would be interesting to compare constitutive Ca²⁺ entry of D76A with EFSAM-6-11.

We have not made this comparison directly in the same experiment. However, since the constitutive puncta are always more pronounced with the D76A mutant, we might expect a larger constitutive Ca²⁺ influx in that case.

Fig.3 e-g: color codes are mixed.

The colors have been corrected.

Reviewer #2 (Remarks to the Author):

Guclur et al report a significant advance in calcium sensing by the STIM proteins, which are ubiquitous ER Ca²⁺ sensors in most animal cells. How STIM1 senses ER Ca levels and transduces Ca concentration changes into store-operated Ca entry is a topic of great intense but remains incompletely understood. Rationalizing that the luminal Ca²⁺ sensing region of STIM1 (EF-SAM) might more faithfully represent the behavior of native protein when it is stably dimerized (mimicking the constitutive dimeric state of STIM1), the group begins their analysis by examining the EF-SAM domain tethered to a soluble dimerizing GrpE domain from Thermus thermophiles. Analysis of Ca²⁺-dependent FRET of the EF-SAM GrpE construct yields the unexpected result that the GrpE fusion proteins of EFSAM display a substantively heightened Ca²⁺ binding sensitivity with a K_d below 10 μM, compared to isolated EF-SAM and STIM1 proteins which were previously reported to respond with K_d values in the 100s of μM.

Critically, ITC and an innovatively designed D4-sensor based fluorescence studies show the presence of multiple binding sites, which are traced to potentially arise from acidic EFSAM surface residues. The study further finds that contrary to current views derived from the isolated EFSAM domains, Ca²⁺ unbinding does not lead to unfolding of its secondary structure, which forces a revision of the prevailing views of Ca²⁺ sensing mechanism of STIM proteins. Instead, it is proposed that Ca²⁺-unbinding leads to a switch in the secondary structure of EFSAM, presumably putting into motion the release of the cytosolic SOAR/CAD domain from CC1. Overall, this paper provides a fresh, thought-provoking perspective on Ca²⁺ binding in the EFSAM domain and STIM1 activation. The experiments are well designed and most conclusions are well supported by the results. However, the paper needs to address some key questions and potential problems as described below. The overall conclusion that EFSAM does not become disordered during Ca²⁺ unbinding is convincing, but the paper could be further improved if some of the surface Ca²⁺ binding sites could be identified and characterized.

Regarding the localization of the Ca²⁺-binding sites, we have made a sustained effort, since 2014, to identify the surface Ca²⁺ binding sites by crystallizing mouse EFSAM-GrpE in the presence of Ca²⁺. These efforts have comprised an initial screen, expanded screens, further screens with crystallization additives, and dehydration of EFSAM-GrpE crystals both prior to shipping to the synchrotron and online at Argonne National Laboratory. There has been a gradual improvement in diffraction, but the best crystals we have obtained so far diffracted to 7-Ångström resolution, which is insufficient to localize bound Ca²⁺. We are now extending the search to EFSAM from other species.

Two years ago we started on the alternative approach of systematic mutagenesis that is reported in the manuscript. We have reached several important conclusions: that EFSAM is structured in the absence of Ca²⁺, that there are multiple Ca²⁺ sites per monomer, and— most importantly— that binding at the EFSAM surface sites is coupled to physiological EFSAM activation. Although crystallization will still be needed to place the Ca²⁺-binding sites in the EFSAM structure, we judge that this part of the story can stand by itself as a contribution to defining the mechanism of STIM Ca²⁺ sensing.

The first half of the paper relies heavily on data generated in the EFSAM-GrpE construct and therefore it is very important to demonstrate that this fusion protein provides a reliable readout for how EFSAM domain behaves in the context of STIM1. How does the dimerization propensity of GrpE compare with that of STIM1? The authors state that

the EFSAM domain behaves differently (i.e. does not unfold as easily in the absence of Ca²⁺) when linked to GrpE. While stabilizing it with GrpE is useful from a technical perspective, could this obscure some conformational changes that occur in STIM1? This is a key point that could derail the conclusions drawn and needs to be further validation beyond the minimal set of controls shown in Figure 1.

These are thoughtful questions, and of course we have considered the same issues. As a broad answer, we would say that the expression of engineered proteins is routine practice in protein chemistry and often a highly productive one, as exemplified, for example, by the structural and early mechanistic insights gained from studies of isolated EFSAM. We would argue that the simplification we have made with EFSAM-GrpE allows us to observe additional detail in the known EFSAM conformational change.

The reviewer's comment embodies several distinct queries:

Re: 'reliable readout'— There are differences, particularly in the Ca²⁺ affinity of the EF-hand site, discussed in detail in our response to Reviewer #1, comment 1. These differences have been valuable and have allowed us to dissect individual features of the STIM1 conformational change and to suggest how EFSAM Ca²⁺ binding interacts with the broader conformational change.

Re: 'dimerization propensity of GrpE'— Both GrpE and SOAR/CAD are stable dimers under physiological conditions.

Re: 'does not unfold as easily in the absence of Ca²⁺'— Not exactly— EFSAM does not unfold in full-length STIM1 either. We have shown this directly in the new experiments described in our response to Reviewer #3, comment 1 (new Figure 6 and new Supplementary Figure 11). This new finding is a concrete demonstration that data from EFSAM-GrpE, judiciously used, can provide insight into full-length STIM1.

Re: 'obscure some conformational changes'— The only conformational change of EFSAM that is documented to occur in full-length STIM1 is the one associated with STIM-STIM FRET. This conformational change is demonstrable in EFSAM-GrpE. We would argue that we have been able to obtain more conformational information than has ever been obtained with full-length STIM1.

Re: 'further validation'— We are limited in that the biophysical or biochemical measurements that have been made on full-length STIM1 are STIM-STIM FRET, crosslinking of TM segments, STIM diffusion, STIM-ORAI FRET, and FRET or co-immunoprecipitation with certain other proteins. Among these, only STIM-STIM FRET provides a relevant comparison for EFSAM-GrpE.

The conclusion that the higher Ca²⁺ concentration dependence of full-length STIM1 relative to the EF-SAM-GrpE construct used here could be due to an "energetic cost exacted by the coupling between Ca²⁺ binding" and ensuing conformational changes is a vague and unsatisfactory explanation for the widely differing Ca²⁺ sensitivities of EF-SAM and STIM1. This critical divergence needs to be more adequately explained. Could the presence of the TM domain between CC1 and the EFSAM in full-length STIM1 offer constraints that lower the Ca²⁺ sensitivity of EFSAM in its native environment? If so, are the conclusions drawn here without the TM domain extendable to full-length STIM1?

We think our views on this issue are in line with those of the reviewer. Constraints such as those imposed by the TM helices, by tethering to the ER membrane, and by the conformational change in CC1-SOAR are exactly the structural constraints that we proposed will lower the Ca²⁺ affinity of EFSAM in the context of full-length STIM1.

As we have detailed at greater length in our response to Reviewer #1, comment 1, the conclusions that we believe extend to STIM1 are that each EFSAM monomer has multiple Ca²⁺-binding sites, and that the luminal domain of STIM1 retains its secondary structure in low Ca²⁺.

Figure 2a,b,d. EFSAM-TM-CC1-GrpE construct was used to parse out the relative contribution of EFSAM domain vs. SOAR/CAD domains to oligomerization observed via FRET. The conclusion that intradimer rearrangement of EFSAM domains accounts for most of the STIM1-STIM1 FRET seen with full-length STIM1 is entirely based on the levels of similar FRET seen between the two constructs....this could be purely correlative and seems like significant over-reach based on simple experiment. More evidence is needed to bolster this strong claim.

Our views are again not far from those of the reviewer. In particular, we did not intend to make the strong claim to which the reviewer objects: Our words were merely that the results ‘suggest that intradimer rearrangement of EFSAM domains accounts for an appreciable part of the STIM1–STIM1 FRET change’. That happens to be our favored interpretation, but it is a suggestion, and readers may accept it or reject it as they choose.

STIM–STIM FRET is not a central point of the manuscript. We included this sidelight because it has been a prevalent view in the STIM–ORAI field that STIM–STIM FRET can be equated with STIM oligomerization, and we felt it was important to note experimental evidence that may point to an alternative interpretation.

Related to the above point, the conclusion that “intradimer rearrangement of EFSAM domains accounts for an appreciable part of the STIM1-STIM1 FRET change” – the data do not rule out that part of the FRET change could arise from oligomerization of CC1 domains downstream of Ca²⁺ unbinding from the EFSAM domains?

Agreed. We have not said that we ruled out a contribution from STIM oligomerization— that remains an area for further research. Whatever the eventual answer, though, it will not change the conclusions of this manuscript regarding EFSAM.

Tamas Balla’s group likewise addressed a closely related question in previous work (PMID: 28724757). This study needs to be cited and discussed in this context.

We inadvertently omitted a citation of this deftly designed study on the cytoplasmic CC1-SOAR interaction and its mechanism of release. We have corrected this omission in the revised manuscript. Because the Korzeniowski *et al* paper focuses on the cytoplasmic CC1-SOAR interaction, it is more directly related to Zhou *et al* 2013, Ma *et al* 2015, and Hirve *et al* 2018, and we have cited its findings in the Hirve *et al* paper.

An important point is to determine how do STIM1-1NQ and STIM1-3NQ mutants affect Ca²⁺ binding to the EFSAM domain? Resolution of this question could be important for the key tenets of this study that Ca binding to the EF hand domain could be stabilized by interaction of EF hand with the SAM domain.

EFSAM-1NQ retained ~5 Ca²⁺ binding sites in the ITC assay, with no indication of an appreciable difference in the titration from that of wildtype EFSAM [**FIGURE R2** for reviewers]. EFSAM-3NQ displayed some aggregation at the high concentrations needed for ITC measurements and so could not be analyzed. Without doubt, there is much further work to be done on crosstalk between the Ca²⁺ sites, and on coupling between the Ca²⁺ sites and EFSAM conformational change. Our conclusions here will serve as a foundation for these further studies, but they fall beyond scope of the current manuscript.

What is the main difference between STIM1-2NQ and STIM1-2NQ (6-11) that switches the protein from a loss-of-function phenotype into a gain-of-function one?

In a few words, STIM1-2NQ fails to form an EFSAM-EFSAM dimer and hence cannot trigger the activating conformational change. The simplest interpretation is that 2NQ substitutions, but only a subset, are detrimental to the dimer interface. ITC shows that EFSAM-2NQ binds Ca²⁺ at its EF-hand, and the ANS fluorescence experiment reports that EFSAM-2NQ undergoes a corresponding Ca²⁺-dependent

conformational change. Therefore the best explanation is that its signalling output is frustrated by a failure of EFSAM-EFSAM dimerization.

In contrast, the 2NQ(6–11) dimer interface is clearly functional, because STIM1-2NQ(6–11) activates upon store depletion. The phenotype of STIM1-2NQ(6–11) can be traced to weaker binding of Ca²⁺ at physiological concentrations to EFSAM, which is reflected in partial activation at resting concentrations of ER Ca²⁺ and enhanced responsiveness to small decreases in ER Ca²⁺.

Figure 5. While the supposition that substantive loss of secondary structure of EFSAM-GrpE should be detectable by CD based on the argument that 50% of the helical content is from EFSAM, this experiment needs a positive control. A comparison within the same experiment to the behavior of EFSAM would be in order here.

To verify that we can observe a loss of EFSAM secondary structure by CD when it occurs, we have followed the thermal denaturation of EFSAM-GrpE. We resolved a clear decrease in α -helical secondary structure of EFSAM-GrpE between 25C and 65C (new Supplementary Figure 8). In contrast, GrpE showed only a minimal decline in α -helical secondary structure in this temperature range, in keeping with published evidence that the main unfolding transition of GrpE occurs above 90C (Nakamura *et al* (2010) J Mol Biol 396, 1000-1011).

Based on supplementary Fig. 6a, does STIM2 EFSAM domain have more Ca²⁺ binding sites (15?) than STIM1?

That is the clear conclusion from Supplementary Figure 6a (Supplementary Figure 7a in the revision), although we would estimate roughly 8–10 sites from the ITC data.

Minor comments:

The text needs to provide greater detail regarding exactly how multiple Ca binding sites are derived from the ITC and the D4 sensor data...it is not be entirely clear to the general reader how this follows from the results shown. Related, the full equations used and the parameters of the fit for Figure 3a, e-h and especially the conclusions from panel "i" need to be more explicitly spelt out.

It is possible to read the approximate number n of binding sites from a 'Wiseman plot' of the ITC data, such as the one in the right-hand panel of Figure 3a. The original paper describing this method of analysis (Wiseman *et al* (1989) Analytical Biochemistry 179, 131–137) showed that under the experimental conditions used in ITC measurements, and for a protein having a single class of binding sites (that is, all the sites have the same K_d and ΔH), n corresponds approximately to the molar ratio of ligand to protein at the midpoint of the titration curve [FIGURE R3 for reviewers]. This feature carries over when there is more than one class of sites, taking the midpoint of the final phase of the titration curve, as illustrated by a collection of Wiseman plots taken from the literature [FIGURES R4 and R5 for reviewers].

Fitting ITC data quantitatively requires adoption of a specific model, such as binding to a single class of sites, binding to two classes of independent sites, or in some cases a more complicated model. We have explained in the text that we did not attempt to fit the EFSAM ITC data to a detailed binding model because of the large number of Ca²⁺-binding sites, the interdependence of binding at the EF-hand and the other Ca²⁺-binding sites, and the coupling of Ca²⁺ binding to a change in intradimer EFSAM–EFSAM interactions. A quantitative description of binding would entail fitting a large number of K_d values, ΔH values, parameters for coupling between the sites, and parameters for coupling between binding at individual sites and the dimer > monomer conformational change. Mathematically, the data do not provide adequate constraints to fit so many parameters.

The standard software supplied with the MicroCal ITC-200 microcalorimeter has options to fit data to models with one or two classes of independent sites. We stress that these models are not appropriate for EFSAM, and that any estimated values for K_d and ΔH are not meaningful. However, the estimate for n may

be credible, because any fit has to regress to the baseline in the region around the true value of n , independent of the model chosen. Using the MicroCal software to fit the data to a model with two classes of independent sites consistently yielded an estimate of 5–6 binding sites per EFSAM monomer.

In the D4 fluorescence competition assay, Ca^{2+} binding to D4 is proportional to the normalized intensity change y . Data points plotted in Figures 3e-3g were normalized as

$$y = (y_{OBS} - y_{MIN}) / (y_{MAX} - y_{MIN}),$$

where y_{MIN} is the intensity measured in Ca^{2+} -free solution, and y_{MAX} the intensity in 25 mM Ca^{2+} . The fitted binding curve in Figure 3e is for binding to a single site with K_d 200 μM ,

$$y = [\text{Ca}^{2+}] / (K_d + [\text{Ca}^{2+}]) \quad \text{[Equation 1]}.$$

The same standard curve is repeated for reference in Figures 3f and 3g.

In Figure 3i, data points from two experiments are plotted individually. Data in the individual experiments were normalized as above, with y_{MIN} the average of the experimental fluorescence intensity values at 5 and 10 nM Ca^{2+} , and y_{MAX} the average of the values at 20 and 25 mM Ca^{2+} . Free Ca^{2+} was estimated for each data point from y and the D4 standard curve with K_d 200 μM , as shown conceptually in Figure 3h. The equation is

$$[\text{Ca}^{2+}]_{\text{FREE}} = K_d (y / (1 - y)),$$

a rearrangement of Equation 1. Bound Ca^{2+} was estimated as

$$[\text{Ca}^{2+}]_{\text{BOUND}} = [\text{Ca}^{2+}]_{\text{TOTAL}} - [\text{Ca}^{2+}]_{\text{FREE}}.$$

Only the data points for total Ca^{2+} 10–500 μM are plotted, because there was no measurable binding at lower total Ca^{2+} concentrations, and the scatter in estimating bound Ca^{2+} by subtracting two large numbers became excessive at higher concentrations.

This description of the analysis has been added to Online Methods.

Related, what happens when the data are fit by fewer than 5 sites? Fits to 1, 2, 3, 4, 5 or more binding sites could be shown to demonstrate the inadequacy of the single or double binding sites.

We have shown fits of raw D4 fluorescence data to models with 1 site and a range of K_d values in Supplementary 2 (Supplementary Figure 3 in the revision). The curves clearly do not replicate the data. Likewise, any curve fitted to the data of Figure 3i based on 1–4 sites would plateau well below the data points. These observations require the conclusion that there is more than one Ca^{2+} -binding site per monomer.

First sentence of “Ca²⁺-dependent conformational change in cells” section is a little ambiguous. It would be helpful to specify that they are referring to the increase in STIM1-STIM1 FRET during store-depletion.

The sentence has been clarified as suggested.

Fig. 3 e-g: Labels and markers don't match.

The mismatch has been corrected.

Sentence describing Fig. 6e on page 14 needs explanation of how the A230C assay works or alternatively, the result can be moved to supplementary information.

An explanation of the assay has been added to the text.

Reviewer #3 (Remarks to the Author):

*STIM1 molecule is essential in store-operated calcium entry process. The previous fundamental view is that STIM1 senses calcium depletion inside endoplasmic reticulum and then aggregates together to become activated, eventually leading to calcium influx through Orai1 channel on plasma membrane. This manuscript by Gudlur et al proposes a novel viewpoint that multiple calcium binding sites interacting with the EF-hand site of EF-SAM domain somehow results in the activation of STIM1 in cells. The major evidence comes from the results that authors designed a new soluble STIM1 construct in which EF-SAM domain coupled through a short linker to GrpE from *Thermus thermophilus*. This new EFSAM-GrpE is soluble and did not adopt drastic conformational changes with or without calcium binding. Interestingly, this soluble protein has 5-6 calcium binding sites. Overall, this paper provided enough evidence to support their viewpoint. And this viewpoint will substantially revise current understanding of calcium sensing by STIM1. However, several critical questions need to be answered before the acceptance of this manuscript.*

1. The manuscript nicely demonstrates that calcium binding induces the subtle conformational changes of EF-SAM domain. However, this conclusion is based on artificial construct EFSAM-GrpE. This result must be confirmed in the full-length STIM1 molecule wild type or near native STIM1 construct (23-444) in vitro. This is very important because different STIM1 constructs or fusion proteins will have different behaviors. Physiologically, full-length STIM1 wild type is the one to carry out function inside cell.

We agree that an important question raised by the EFSAM-GrpE observations is whether EFSAM in full-length STIM1 is obligated to unfold when it transitions to the active dimeric form. This question posed an experimental challenge, because the biophysical methods that we used to probe EFSAM structure in the manuscript were not readily applicable to STIM1 in ER membranes; and because the alternative of purifying full-length STIM1, reconstituting it into membranes, and demonstrating normal physiological function before undertaking the actual experiment would have been a nontrivial and lengthy exercise. A practical approach was to probe whether cysteine residues engineered at sites that are buried or partially buried in Ca²⁺-bound EFSAM-GrpE become exposed (or not) upon dissociation of Ca²⁺ from full-length STIM1 in ER membranes.

We engineered a panel of constructs appropriate for the experiment, and carried out the experiment in parallel with full-length STIM1 in ER membranes and with EFSAM-GrpE. Except for one M>C replacement, the engineered proteins had relatively conservative V>C, T>C, and S>C replacements at sites that are buried or partially buried in the NMR structure. All the mutant STIM1 proteins (expressed as GFP-STIM1) responded normally to ER store depletion. We prepared membranes in the same way as for the STIM1-STIM1 crosslinking experiments in which we have shown that STIM1 undergoes a conformational change in response to Ca²⁺ (Hirve *et al* (2018) and this manuscript), incubated the membranes in buffer with 0.5 mM EGTA or 2 mM Ca²⁺, and probed exposure of the individual engineered cysteine residues by reaction with biotin-maleimide.

The residues tested showed similar low accessibility in the presence of Ca²⁺ for full-length STIM1 and for EFSAM-GrpE. The exposed S58C control was well labelled, and the buried cysteine residues in EFSAM-GrpE became exposed and could be labelled upon denaturation with 6M urea. However, only minor changes in accessibility of the buried residues were seen in the two proteins when comparing the Ca²⁺ and EGTA conditions. Thus the EFSAM domain of full-length STIM1 did not unfold upon dissociation of Ca²⁺. The results are included in the revised manuscript as new Figure 6 and new Supplementary Figure 11.

2. The authors used ITC and fluorescence assay to show that calcium binds to multiple sites of EF-SAM domain. EFSAM-GrpE is a dimer and contains at least two calcium binding EF-hands. It is easy to understand that two EF-hands may have cooperative binding. This explains the findings from Figure 3a. Thus, multiple calcium binding sites conclusion is not suitable for ITC experiment. The next important question is that where are those binding sites? Which specific residues they are.

The EFSAM-GrpE concentrations stated in the figures are monomer concentrations. Hence there is only one Ca²⁺ binding EF-hand site per EFSAM-GrpE monomer, and the additional Ca²⁺ binding must occur at a different class of sites. We have revised the legend to Figure 3a to ensure that this is clear to readers.

We agree that identification of the additional Ca²⁺-binding sites in the EFSAM structure is a central question, and our efforts to define the sites were described above in a response to Reviewer #2.

3. It is puzzling that STIM1-2NQ mutant (D77N, D82N, E86Q, D89N, E90Q, E94Q, D100N, E111Q, D112N, E118Q, and D119N) is inactive from the Figure 4b and contains one calcium binding site from the Figure 4c, however, STIM1-2NQ(6-11) (E94, D100, E111, D112, E118, and D119) is self-active from Figure 6a,c and contains multiple calcium binding sites from Figure 6d. These results seem to be contradictory. The authors might address the reason behind these inconsistent results.

The results are not inconsistent, but, as the reviewer points out, they require careful attention. Both proteins contain a functional EF-hand site and so are responsive to Ca²⁺— the ANS fluorescence change (Supplementary Figure 10f in the revised manuscript) establishes this point for EFSAM-2NQ. As we see it, the broad region 2 surface of EFSAM supports two separate functions, one of which is the dimerization discussed above, the other Ca²⁺ binding that is cooperative with Ca²⁺ binding at the EF-hand site. The 2NQ mutations abrogate both functions, so that, despite Ca²⁺ dissociation from the EF-hand, there is no dimerization. The 2NQ(6–11) mutations affect only the concentration dependence of Ca²⁺ binding, and do not prevent EFSAM-EFSAM dimerization. See also our response to a different version of this question posed by Reviewer #2.

4. back to fundamental question. If the EF-SAM does not adopt drastic conformational changes upon calcium loss, what force drives the conformational changes of CC1 region to expose SOAR or CAD domain. Where does the energy come from? How to explain the characteristic feature of STIM1 punta upon activation ?

This is a crucial point for understanding the coupling between Ca²⁺ dissociation and STIM1 activation. The energy comes from the favorable EFSAM-EFSAM interaction between Ca²⁺-free EFSAM domains. Apparently only a modest rearrangement of secondary structure within each domain is needed to allow this intramolecular dimerization. We have explained the mechanism for transmission of the EFSAM-EFSAM dimerization signal to the cytoplasmic domain in Hirve *et al* 2018. As regards formation of STIM puncta, release of the polybasic tail and the SOAR/CAD domain targets STIM to puncta— this part of the mechanism is well established— via STIM polybasic tail–lipid and SOAR/CAD–ORAI interactions.

FIGURE R1 EFSAM-2NQ(6-11) / CD spectra and ANS fluorescence.

FIGURE R2 Representative ITC data for EFSAM-1NQ.

FIGURE R3 Calculated 'Wiseman plots' for a protein with n identical sites, at different protein concentrations specified as different c values. Recommended c values are in the range 10–500. From Wiseman *et al* (1989) Analytical Biochemistry 179, 131-137.

FIG. 3. Simulated binding isotherms for various values of the parameter c (equal to the product of the binding constant times the total macromolecule concentration), presented in derivative format. See text for details.

FIGURE R4 Examples of Wiseman plots for cases with one or two classes of sites and $n = 2$. From Freire *et al* (2009) *Methods Enzymol* 455, 127-155.

FIGURE R5 Example of a Wiseman plot for a case with three classes of sites and $n = 4$. From Le *et al* (2013) *Analytical Biochemistry* 434, 233-241.

Reviewers' Comments:

Reviewer #1:

Remarks to the Author:

No further comments.

Reviewer #2:

Remarks to the Author:

Hogan and colleagues have addressed most of the concerns that I raised in the previous submission. In general, there wasn't that much new data, mostly additional clarification/reasoning about why the existing data answers the questions that they are addressing. As expected, all three reviewers wanted some more validation and explanation of how GrpE is a faithful surrogate for STIM1. The authors argue that many experiments suggested by the reviewers on this issue are not easily feasible or out of scope, that may be so, and I think this paper is probably solid enough as is. The newly added explanation of how binding sites can be calculated from ITC curves in the Online Methods is helpful. It would be nice to elaborate on how the additional Ca²⁺ binding sites on the EFSAM domain on STIM2 influences its physiological function. At the moment, the STIM2 data is basically just there with no explanation in the main text at all, which is a little strange.

Reviewer #3:

Remarks to the Author:

The authors fully addressed my questions and thus I recommend to accept this manuscript.

POINT-BY-POINT RESPONSE TO REVIEWERS' CONCERNS

Reviewer #1 (Remarks to the Author):

No further comments.

Reviewer #2 (Remarks to the Author):

Hogan and colleagues have addressed most of the concerns that I raised in the previous submission. In general, there wasn't that much new data, mostly additional clarification/reasoning about why the existing data answers the questions that they are addressing. As expected, all three reviewers wanted some more validation and explanation of how GrpE is a faithful surrogate for STIM1. The authors argue that many experiments suggested by the reviewers on this issue are not easily feasible or out of scope, that may be so, and I think this paper is probably solid enough as is. The newly added explanation of how binding sites can be calculated from ITC curves in the Online Methods is helpful. It would be nice to elaborate on how the additional Ca²⁺ binding sites on the EFSAM domain on STIM2 influences its physiological function. At the moment, the STIM2 data is basically just there with no explanation in the main text at all, which is a little strange.

Like Reviewer #2, we believe that a thorough analysis of STIM2 activation would be worthwhile. Co-author Youjun Wang, in particular, is following this track. The analysis in this paper has been more limited: We have shown only that STIM2 EFSAM domain has multiple Ca²⁺-binding sites, and that the 2NQ replacements prevent activation in STIM2 as they do in STIM1.

We do not feel able to elaborate in the main text on how the multiple STIM2 luminal Ca²⁺-binding sites relate to STIM2 physiological function, for two reasons. First, the data on STIM2 did not prove to be central to the paper, and STIM2 is not mentioned in the main text. Second, we did not test experimentally whether the additional STIM2 EFSAM sites interact with the EF-hand site to tune physiological activation, as they do in STIM1. We share the Reviewer's interest in the question, but we are not comfortable going beyond the interpretation of the limited experimental data that we have given in Supplementary Note 2.

Reviewer #3 (Remarks to the Author):

The authors fully addressed my questions and thus I recommend to accept this manuscript.